# TRIM21-mediated proteasomal degradation of SAMHD1 regulates its antiviral activity

Zhaolong Li[1], Chen Huan[1], Hong Wang[1], Yue Liu[1], Xin Liu[1], Xing Su[1], Jinghua Yu[1], Zhilei Zhao[1], Xiao-Fang Yu[1,2], Baisong Zheng[1] & Wenyan Zhang[1,*] (iD)

## Abstract

**SAMHD1 possesses multiple functions, but whether cellular factors regulate SAMHD1 expression or its function remains not well characterized. Here, by investigating why cultured RD and HEK293T cells show different sensitivity to enterovirus 71 (EV71) infection, we demonstrate that SAMHD1 is a restriction factor for EV71. Importantly, we identify TRIM21, an E3 ubiquitin ligase, as a key regulator of SAMHD1, which specifically interacts and degrades SAMHD1 through the proteasomal pathway. However, TRIM21 has no effect on EV71 replication itself. Moreover, we prove that interferon production stimulated by EV71 infection induces increased TRIM21 and SAMHD1 expression, whereas increasing TRIM21 overrides SAMHD1 inhibition of EV71 in cells and in a neonatal mouse model. TRIM21-mediated degradation of SAMHD1 also affects SAMHD1-dependent restriction of HIV-1 and the regulation of interferon production. We further identify the functional domains in TRIM21 required for SAMHD1 binding and the ubiquitination site K622 in SAMHD1 and show that phosphorylation of SAMHD1 at T592 also blocks EV71 restriction. Our findings illuminate how EV71 overcomes SAMHD1 inhibition via the upregulation of TRIM21.**

**Keywords** EV71 infection; interferon induction; regulation; SAMHD1 inhibition; ubiquitin–proteasome degradation

**Subject Categories** Immunology; Microbiology, Virology & Host Pathogen Interaction; Post-translational Modifications & Proteolysis

## Introduction

Sterile alpha motif and histidine–aspartic acid domain-containing protein 1 (SAMHD1) was initially identified as an ortholog of the mouse interferon-γ-induced gene MG11 and has been extensively investigated since its identification as an anti-HIV-1 restriction factor in 2011 [1–3]. SAMHD1 has been reported as an effector of innate immunity, a restriction factor for retroviruses, and a regulator of the cell cycle, of retrotransposons, and of autoimmunity, among other functions [4–11]. The SAMHD1-mediated restriction of retroviruses and several DNA viruses was demonstrated to be achieved mainly through depleting the intracellular pool of deoxynucleotide triphosphates (dNTPs) to levels that are insufficient to support viral DNA synthesis [6,9,12–14], although several reports argued that nuclease activity contributed to its antiviral activity [15–17]. SAMHD1 is also a nucleic acid binding protein that binds DNA and DNA:RNA duplexes [18–20]. A more complicated finding was that when SAMHD1 is phosphorylated at T592 by CDK/cyclin D/E in G1-like phase, it loses its HIV restriction activity [13,21]. Recently, SAMHD1 was found to function in DNA damage repair, which broadens its known cellular functionality [5,22].

To escape restriction by SAMHD1, HIV-2 and SIV evolved the Vpx protein to degrade SAMHD1 via the proteasome pathway by hijacking the Cul4A/DDB1/DCAF1 E3 ligase [1,2,23,24]. The Ratner team reported that cyclin L2 causes SAMHD1 degradation by interacting with DCAF1 in a proteasome-dependent manner and is required for HIV-1 replication in macrophages [25].

TRIM21 is a member of the tripartite motif (TRIM) family, which was reported to be a cytosolic Fc receptor required for antibody-dependent intracellular neutralization and intracellular antibody-mediated degradation [26,27] that restricts pathogens through the proteasome pathway [28,29]. TRIM21 (Ro52) was initially identified as an antibody-binding protein by yeast two-hybrid analysis [30], and subsequent reports identified the interaction region as the carboxyl-terminal B30.2 domain of TRIM21, which binds to residues on Fc at the CH2–CH3 domain interface [31–33]. A recent report suggested that TRIM21 can be used as a tool to shut down proteins in different cells by utilizing its antibody binding property and the cellular protein degradation machinery [34]. In addition, accumulating evidence showed that TRIM21 also facilitates proteasomal degradation of virions and influences innate immune responses [28,29,35–40]. Therefore, the role of TRIM21 in viral infection is more complicated due to its E3 ligase function.

In contrast to the inhibitory effects of SAMHD1 against retroviruses and DNA viruses [16, 41–45], the impact of SAMHD1 on other RNA viruses such as enteroviruses (EVs) has not been previously investigated. It is well known that enterovirus 71 (EV71) is a

---

1 The First Hospital of Jilin University, Institute of Virology and AIDS Research, Changchun, China
2 Cancer Institute (Key Laboratory of Cancer Prevention and Intervention, Ministry of Education), Second Affiliated Hospital, School of Medicine, Zhejiang University, Hangzhou, China
*Corresponding author. Tel: +86 431-88782148; Fax: +86 431-85654528; E-mail: zhangwenyan@jlu.edu.cn

major pathogen that causes hand-foot-and-mouth disease (HFMD), which is a severe public problem threatening children under 5 years [46,47]. In this study, we demonstrate that SAMHD1 is a restrictive factor for EV71. However, EV71 overcomes the inhibitory effect of SAMHD1 by upregulating TRIM21 expression, which interacts and degrades SAMHD1 through the proteasomal pathway. Furthermore, we show that the PRY and SPRY domains of TRIM21 are required for SAMHD1 binding, and K622 of SAMHD1 is the ubiquitination site for TRIM21. Interestingly, we find that SAMHD1 restriction is independent of its dNTPase or nuclease activities, but closely related to its phosphorylation. SAMHD1 phosphorylated at T592 loses the ability to restrict EV71 replication, which is similar to HIV restriction. These findings broaden our understanding of SAMHD1 function and of the host regulation of SAMHD1 and provide a stratagem to regulate SAMHD1 function.

## Results

### SAMHD1 is responsible for the different sensitivity of RD and HEK293T cells to EV71

Human rhabdomyosarcoma RD cells (RD) and HEK293T cells were infected with equal amounts of EV71 viruses at a multiplicity of infection (MOI) of 0.1, and different susceptibility of the two cell lines to EV71 was observed, suggesting that HEK293T cells might contain factors that inhibit EV71 replication (Fig 1A and B). Host restriction factors such as APOBEC3 family members, BST-2, SAMHD1, Mx2, GBP5, and SERINC5 have been reported to be the first lines of defense against viral invasion [1,2,48–51]. Therefore, we first investigated the mRNA levels of the above-mentioned host factors in these two cells. The results showed that the mRNA level of SAMHD1 was far higher than that of BST-2 and other host factors (Fig 1C). By immunoblotting analysis, we further observed that protein level of SAMHD1 in HEK293T cells was about fivefold higher than that in RD cells, while BST-2 expression was not detected in either HEK293T or

RD cells, in contrast to HepG2 cells (Fig 1D), which were reported as BST-2-positive cells [52]. Thus, we hypothesized that SAMHD1 might play a critical role in EV71 restriction and examined the effect of SAMHD1 on EV71 replication.

To investigate whether SAMHD1 is a major restrictive factor against EV71 replication and affects the different replicative dynamics of EV71 in the two cell lines, HEK293T SAMHD1 knockdown cells were constructed and then infected with EV71 at a MOI of 0.1. At different time points post-infection, we found that EV71 VP1 expression in HEK293T-shSAMHD1 cells and supernatants was higher than those in HEK293T-pLKO.1 control cells and supernatants at 48 h post-infection (Fig 1E), indicating that SAMHD1 inhibits EV71 replication. EV71 RNA analysis and virus titer detection also showed that SAMHD1 inhibited EV71 replication (Fig 1F and G). Although SAMHD1 is expressed at lower levels in RD cells, knockdown of SAMHD1 still permitted faster replication of EV71 compared to RD-pLKO.1 control cells (Fig 1H–J). In previous studies, SIV Vpx was demonstrated to degrade SAMHD1 via the proteasome pathway [1,2]. We transfected the Vpx expression vector into HEK293T cells to overcome the restriction of EV71 replication by SAMHD1. As expected, Vpx-mediated SAMHD1 degradation enhanced EV71 replication, as showed by VP1 expression, EV71 RNA level, and virus titer detection (Fig 1K–M). Taken together, these data demonstrate that SAMHD1 is a major restriction factor for EV71 replication.

### E3 ligase TRIM21 is a regulator of SAMHD1 in RD cells

To further verify the restriction of EV71 replication by SAMHD1, we stably expressed SAMHD1 hemagglutinin (HA) in both RD and HEK293T cells by using a lentivirus system. Surprisingly, we were not able to detect exogenous expression of SAMHD1-HA in RD cells relative to HEK293T cells (Fig 2A). Therefore, we speculated that factors in RD cells reduce the expression of SAMHD1-HA. Interestingly, the expression of SAMHD1-HA was rescued after treatment with the proteasomal inhibitor MG132, demonstrating our hypothesis that there is an E3 ligase system in RD cells responsible for the

---

**Figure 1. SAMHD1 suppresses EV71 replication.**

A–D The expression of SAMHD1 is negatively correlated with EV71 viral replication ability. HEK293T and RD cells were infected with EV71 at a MOI of 0.1; then, the cells and supernatants were harvested at the indicated time points. (A) Immunoblotting (IB) analysis of EV71 VP1 in cells, with tubulin as a loading control. EV71-VP1 protein in the supernatants was detected after ultracentrifugation. (B) EV71 viral RNA levels in cell lysates and supernatants were detected by RT–qPCR with GAPDH as a control. ($n$ = 3, mean ± SD, ns stands for no significance, paired $t$-test) (C) The mRNA levels of the host restriction factors were detected by RT–qPCR in infected or uninfected HEK293T or RD cells, and the expression levels of the target genes were normalized to GAPDH. ($n$ = 3, mean ± SD, ns stands for no significance, paired $t$-test). (D) IB analysis of SAMHD1 and BST-2 protein levels with tubulin as a loading control. The densities of bands were analyzed with ImageJ software to calculate the values relative to that of tubulin.

E–J SAMHD1 knockdown enhances EV71 replication. The stable cell lines pLKO.1 and sh-SAMHD1 constructed in HEK293T (E) and RD (H) cells were infected with EV71 at a MOI of 0.1 and 0.05, respectively, and cells and supernatants were harvested at the indicated time points. IB analysis of EV71 VP1 and SAMHD1 in cells was performed with tubulin as a loading control. EV71-VP1 protein in the supernatants was detected after ultracentrifugation. (F and I) EV71 viral RNA in cell lysates was detected by RT–qPCR with GAPDH as a control ($n$ = 3, mean ± SD, *$P$ < 0.05, **$P$ < 0.01, paired $t$-test). (G and J) Viral titers in the supernatants were measured by the cytopathic effect method. The results represent the means ± SD from three independent experiments. Statistical significance was analyzed using Student's $t$-test (*$P$ < 0.05, **$P$ < 0.01).

K HEK293T cells transfected with VR1012, which is a eukaryotic expression vector, or Vpx-HA were infected with EV71 at a MOI of 0.1, and then, cells and supernatants were harvested at the indicated time points. IB analysis of EV71-VP1, SAMHD1, and Vpx-HA in cell lysates was performed with tubulin as a loading control. EV71-VP1 was detected in the supernatants after ultracentrifugation.

L EV71 viral RNA was detected in cell lysates by RT–qPCR with GAPDH as a control ($n$ = 3, mean ± SD, **$P$ < 0.01, paired $t$-test).

M Viral titers in the supernatants were measured by the cytopathic effect method. The results represent the means ± SD from three independent experiments. Statistical significance was analyzed using Student's $t$-test (**$P$ < 0.01).

Source data are available online for this figure.

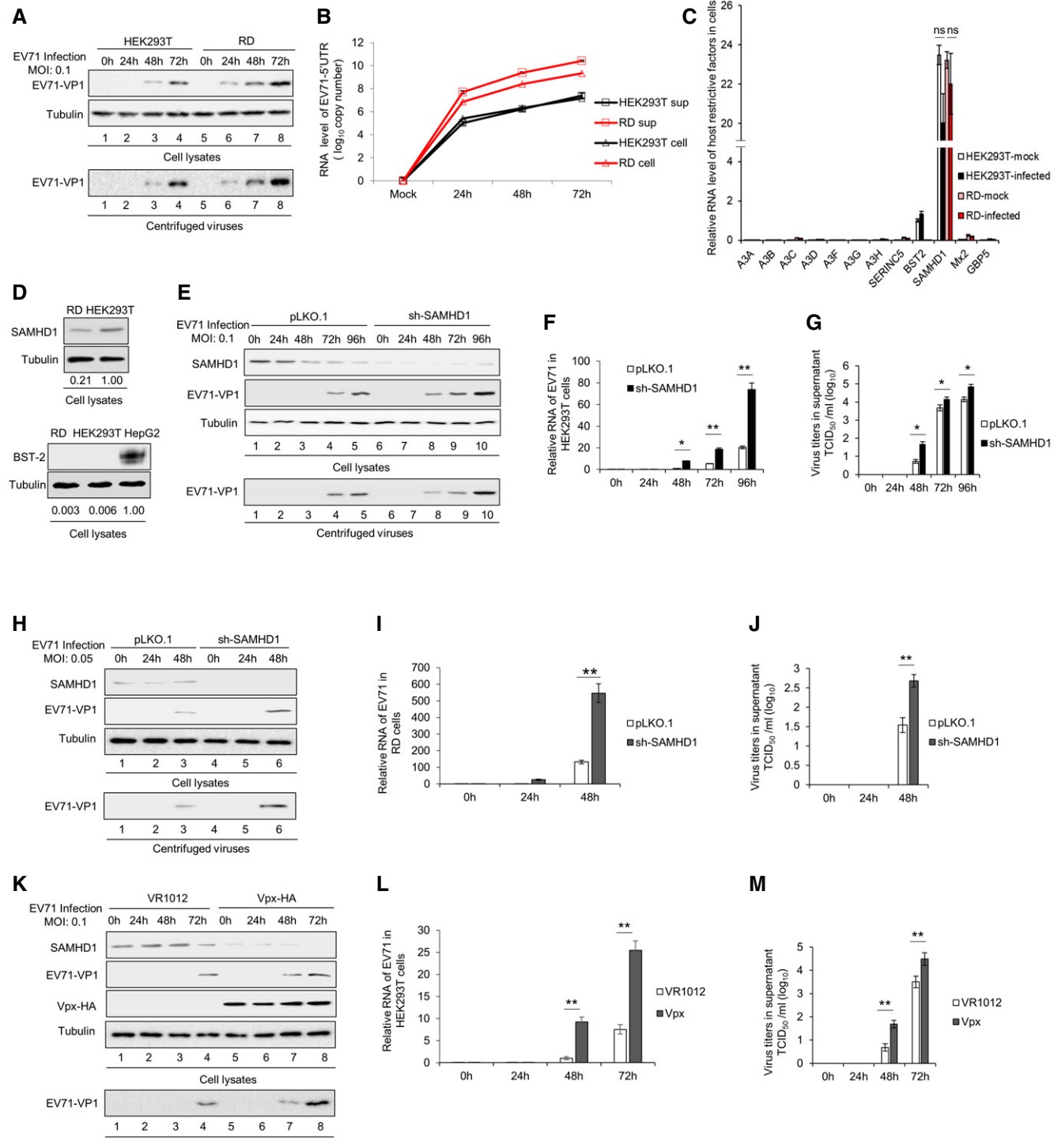

**Figure 1.**

degradation of SAMHD1 (Fig 2B). To identify the E3 ligase complex, we performed a co-immunoprecipitation (co-IP) assay (Fig 2C) and analyzed the eluent by mass spectrometry. From the proteins identified, we selected TRIM21, an E3 ubiquitin ligase, for further investigation (Appendix Table S1).

Overexrpession of TRIM21 indeed induced the degradation of SAMHD1, but MG132 rescued the decreased expression of SAMHD1

(Fig 2D), while another TRIM family member, the ubiquitin E3 ligase TRIM25, could not degrade SAMHD1 (Fig 2E), indicating that TRIM21 specifically recognizes and degrades SAMHD1. Reverse co-IP assays further verified that SAMHD1 was detected in TRIM21 immunoprecipitates (Fig 2F). TRIM21 was previously identified as an intracellular Fc receptor [53,54]. In order to exclude nonspecific interactions caused by TRIM21 binding to the antibody, we used the

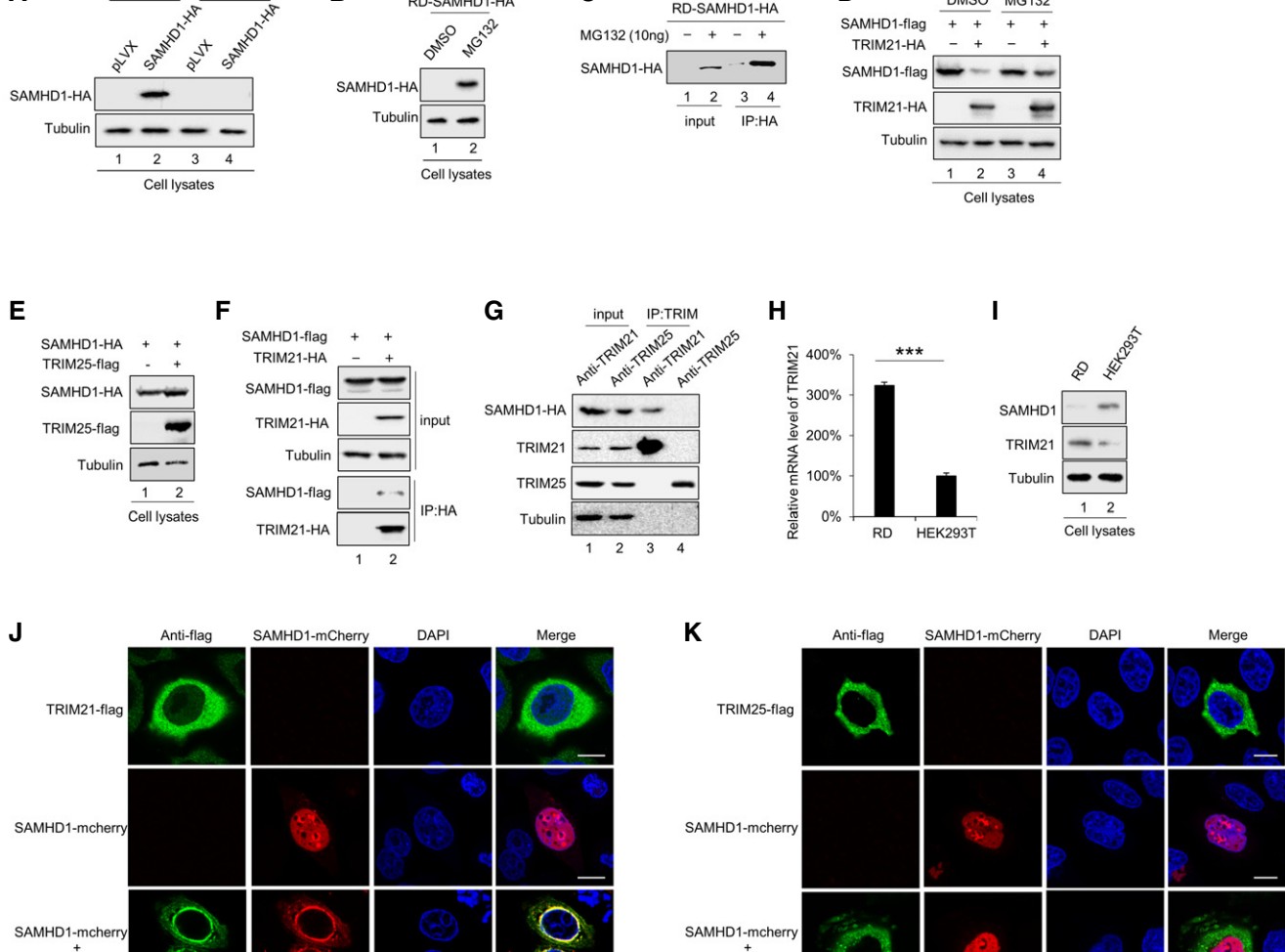

**Figure 2. SAMHD1 is degraded by TRIM21 via the proteasomal pathway.**

A     Stable SAMHD1-HA-overexpressing cell lines were constructed in HEK293T and RD cells and detected by IB with pLVX as a negative control.
B     RD SAMHD1-HA cell lines were treated with dimethyl sulfoxide (DMSO) or 10 μM MG132 for 12 h prior to harvest and subjected to IB with tubulin as a loading control.
C     RD SAMHD1-HA cell lines were treated as in (B), and then, HA immunoprecipitation (IP) and IB were performed. The IP elution was analyzed by mass spectrometry.
D, E  TRIM21 (D) but not TRIM25 (E) induced the degradation of SAMHD1, and MG132 rescued the TRIM21-mediated degradation of SAMHD1. HEK293T cells were cotransfected with SAMHD1 and VR1012 or TRIM21 or TRIM25 expression plasmids with or without MG132 treatment for 12 h prior to harvest and then subjected to IB with tubulin as a loading control.
F     Reverse co-IP confirmed the interaction between SAMHD1 and TRIM21. SAMHD1-flag was transfected with TRIM21-HA into HEK293T cells, and then, the cells were treated as in (B) and subjected to HA IP and IB.
G     Endogenous TRIM21 but not TRIM25 still interacted with SAMHD1. RD-SAMHD1-HA cells treated as in (B) were subjected to TRIM21 or TRIM25 IP and IB.
H, I  (H) mRNA (n = 3, mean ± SD, ***P < 0.001, paired t-test) and (I) protein levels of TRIM21 in RD and HEK293T cells.
J, K  TRIM21 (J) but not TRIM25 (K) colocalized with SAMHD1. Images were taken under a Zeiss LZM710 confocal microscope, Bars, 10 μm.

Source data are available online for this figure.

TRIM antibody to precipitate endogenous TRIM and could confirm the specific binding of TRIM21, but not TRIM25 to SAMHD1 (Fig 2G). By qPCR and Western blot analysis, we found higher RNA and protein levels of TRIM21 in RD cells than in HEK293T cells, which is consistent with the lower expression of SAMHD1 in RD cells (Fig 2H and I). A confocal assay also demonstrated that SAMHD1 completely translocated from the nucleus to the nuclear membrane, where it colocalized with TRIM21 but not TRIM25 (Fig 2J and K).

**TRIM21 promotes EV71 replication by degrading SAMHD1**

To clarify the role of TRIM21 in SAMHD1-mediated restriction on EV71 replication, we constructed stable TRIM21 knockdown RD

cells using shRNA (Fig EV1A) or full knockout cells using the CRISP-Cas9 technology (CRISP-Cas9-TRIM21, Fig 3A) and observed a corresponding increase in SAMHD1 expression compared to that in negative control cells. The negative control cells and RD-sh-TRIM21-3 or CRISP-Cas9-TRIM21 cells were then infected with EV71 at a MOI of 0.05 for 48 h. By immunoblotting, and EV71 RNA and EV71 virus titer detection, we found that TRIM21 knockdown or knockout led to faster replication and higher titers of EV71 (Figs EV1B and C, and 3B–D). Interestingly, we observed that the upregulated SAMHD1 resulting from TRIM21 knockdown or knockout gradually decreased with EV71 replication, whereas TRIM21 expression gradually increased with EV71 replication, indicating that EV71 gradually overcomes the restriction effect of SAMHD1 via upregulating TRIM21 expression (Figs EV1B and C, and 3B and C). In order to substantiate the relevance *ex vivo*, peripheral blood mononuclear cells (PBMC) from three healthy donors were employed to examine EV71 replication and SAMHD1 expression in TRIM21 knockdown cells. The results showed that TRIM21 knockdown rescued SAMHD1 expression (Fig EV1D, Lanes 4, 8, and 12 compared to 3, 7, and 11), which correlates with decreased EV71 viral RNA levels (Fig EV1E). The opposite effect was observed in HEK293T cells transfected with TRIM21-HA, which resulted in

stronger EV71 replication, further supporting the idea that TRIM21-mediated degradation of SAMHD1 enabled EV71 replication (Fig 3E and F). These results demonstrated that TRIM21-induced degradation of SAMHD1 permitted faster replication of EV71. Some TRIM family members have been implicated in the modulation of innate immunity and antiviral activity [36,38–40,55]. Here, we demonstrate that overexpression of TRIM21 led to increased EV71 replication in a SAMHD1-dependent manner, because TRIM21 lost its effect on EV71 in sh-SAMHD1 cells (Fig 3G and H). In order to verify the relationship between SAMHD1 and TRIM21 during EV71 infection, we also detected EV71 replication in RD cells with TRIM21 and SAMHD1 individually depleted, as well as in combination. We found that EV71 replication was negatively correlated with SAMHD1 levels (Fig EV2A and B). More than that, upon the treatment of TRIM21-depleted cells with MG132, we found that MG132 had no effect on SAMHD1 stability (Fig EV2C). Interestingly, we observed that knockdown of SAMHD1 had no effect on the replication of respiratory syncytial virus (RSV) during multiple infections (Fig EV2D–E), but obviously inhibited the replication of vesicular stomatitis virus (VSV) within 24 h (Fig EV2F). Similar to the effect on EV71, overexpression of TRIM21 increased VSV replication in a SAMHD1-dependent manner (Fig EV2F).

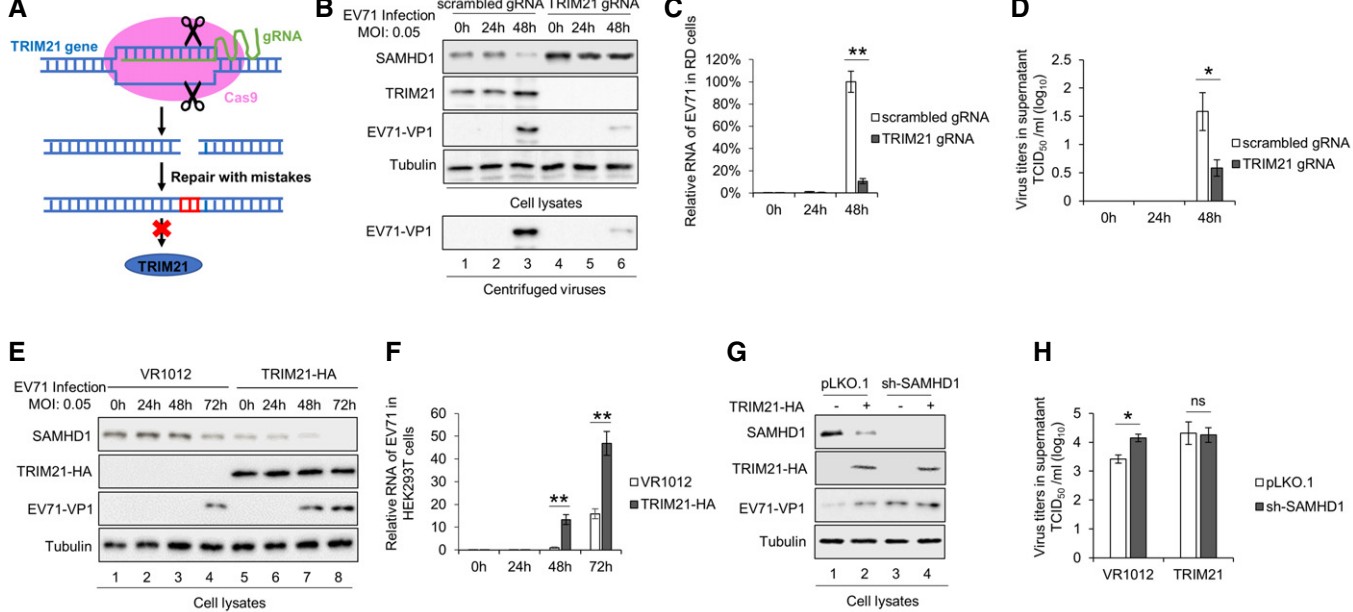

**Figure 3. The degradation of SAMHD1 mediated by TRIM21 affects anti-EV71 restriction of SAMHD1.**

A–D  TRIM21 knockout in RD cells released SAMHD1 inhibition on EV71 replication. (A) Illustration of gRNA-guided TRIM21 knockout in cells. (B) Negative control or RD-CRISPR-Cas9-TRIM21 RD cells were infected with EV71 at a MOI of 0.05 for the indicated time and harvested for SAMHD1, TRIM21, and EV71-VP1 detection by IB. Tubulin served as a loading control. (C) EV71 RNA levels in (B) were detected by RT–qPCR with GAPDH as a control (*n* = 3, mean ± SD, **P < 0.01, paired *t*-test). (D) Viral titers in the supernatant were measured by the cytopathic effect method. The results represent the means ± SD from three independent experiments. Statistical significance was analyzed using Student's *t*-test (**P < 0.01).

E, F  Overexpression of TRIM21 in HEK293T cells increased EV71 replication by degrading SAMHD1. HEK293T cells were transfected with VR1012 or TRIM21-HA for 24 h and then infected with EV71 at a MOI of 0.05. (E) Cells were harvested at the indicated time points and subjected to IB with tubulin as a loading control. (F) EV71 viral RNA in cell lysates was detected by RT–qPCR with GAPDH as a control (*n* = 3, mean ± SD, **P < 0.01, paired *t*-test).

G, H  TRIM21 itself has no effect on EV71 replication. VR1012 or TRIM21 was transfected into pLKO.1 or sh-SAMHD1 HEK293T cells for 24 h and infected with EV71. (G) Cells were harvested and subjected to IB analysis. (H) Viral titers in the supernatants were measured by the cytopathic effect method. The results represent the means ± SD from three independent experiments. Statistical significance was analyzed using Student's *t*-test (*P < 0.05).

Source data are available online for this figure.

## Interferon production induced by EV71 infection upregulates TRIM21 expression

Most TRIM family members are interferon (IFN)-induced proteins [38,56]. We thus hypothesized that EV71 infection would stimulate innate immunity responses such as IFN production, which will cause TRIM21 upregulation. As expected, we indeed observed that type I IFN-α and IFN-β were stimulated by EV71 infection over time in both RD and HEK293T cells (Fig EV3A). At the same time, the mRNA levels of SAMHD1 and TRIM21 also were gradually upregulated (Fig EV3B). Consistent with previous reports [57], we observed that IFN-α induced the expression of TRIM21, even in sh-TRIM21 cells (Fig EV3C). Moreover, knockdown of IFN alpha and beta receptor subunit 1 (IFNAR1) (Fig EV3D) abolished the upregulation of TRIM21 upon EV71 infection (Fig EV3E Lanes 2 and 4, and 6 and 8), indicating that increased TRIM21 caused by EV71 infection is IFN-dependent. Knockdown of IFNR1 only slightly increased EV71 replication due to increased SAMHD1 (Fig EV3E Lanes 6 compared with Lane 2), while knockdown of both SAMHD1 and IFNAR1 moderately increased EV71 replication (Fig EV3E Lane 8 compared with Lane 2). We also observed that increased SAMHD1 did not totally overwhelm EV71 replication, suggesting that IFN production involves various antiviral pathways important for EV71

inhibition. In addition, as there was no effect of EV71 nonstructural proteins on SAMHD1 expression, it seems that TRIM21 is a natural negative feedback mechanism to limit SAMHD1 inhibition after IFN response (Fig EV3F).

## EV71 infection is positively and negatively correlated with the expression of TRIM21 and SAMHD1 *in vivo*, respectively

Neonatal mouse models have been employed to evaluate EV71 infection *in vivo*, and EV71 infection in these causes severe lesions in muscle tissues but not heart or lung tissues due to the strong tissue tropism of EV71, although the exact mechanism is undetermined until now [58–60]. To better understand the correlation of the expression of TRIM21 and SAMHD1 with susceptibility to EV71 *in vivo*, we examined the mRNA and protein levels of TRIM21 and SAMHD1 in different tissues of neonatal mice and found that mRNA and protein levels of SAMHD1 in heart and lung tissues were higher than in spinal and hind leg muscles, whereas the mRNA and protein levels of TRIM21 in these tissues showed opposite tendencies (Fig 4A and B). In addition, by quantifying SAMHD1 levels by immunohistochemical (IHC) staining, we found that heart and lung tissues contain higher amounts of SAMHD1, followed by spinal muscles and hind leg muscles, which is

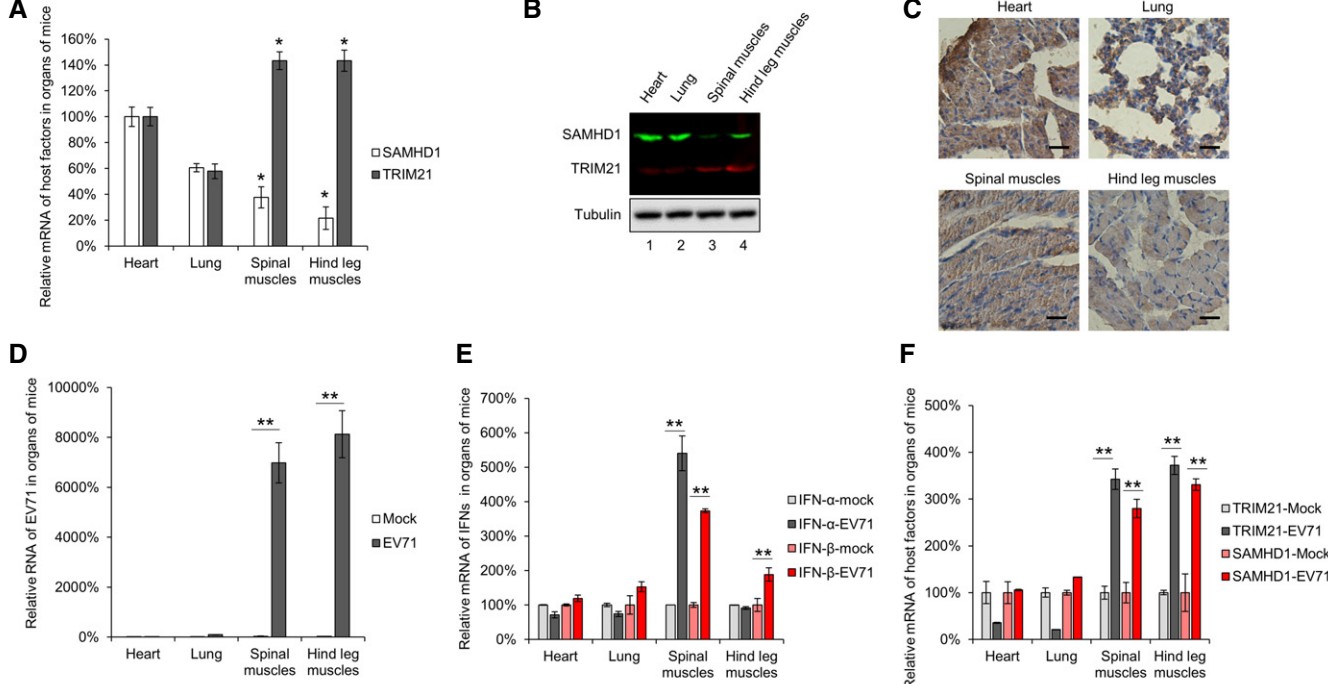

**Figure 4. The tissue tropism upon EV71 infection in neonatal mouse model is positively correlated with TRIM21 and negatively correlated with SAMHD1.**

A   The mRNA levels of SAMHD1 and TRIM21 in different tissues of neonatal mice were detected by RT–qPCR with GAPDH as a control, and the expression of SAMHD1 and TRIM21 in the heart was set as 100% (*n* = 3, mean ± SD, **P* < 0.05, paired *t*-test).

B   The protein levels of SAMHD1 and TRIM21 in different tissues of mice were examined by near-infrared spectrum (NIR) Western blot with tubulin as a control.

C   Immunohistochemical (IHC) staining results of SAMHD1 in various tissues of mice. Bars, 100 μm.

D–F The mRNA levels of EV71 virus and host factors in mock- or EV71-infected mice were detected by RT–qPCR with GAPDH as a control. The expression of EV71 (D), IFN-α and IFN-β (E), and SAMHD1 and TRIM21 (F) were presented, and the levels of corresponding gene in mock-infected mice were set as 100% (*n* = 3, mean ± SD, ***P* < 0.01, paired *t*-test).

Source data are available online for this figure.

consistent with the findings in Fig 4A and B. This suggested that the different expression levels of TRIM21 and SAMHD1 in various mouse tissues are the major reasons why EV71 has a strong tissue tropism. Then, we further investigated whether the expression of TRIM21 and SAMHD1 was associated with the susceptibility to EV71. After infection with EV71, spinal muscles and hind leg muscles of mice, which express relative low level of SAMHD1 (Fig 4A–C), had higher levels of EV71 (> 60-folds), whereas there was barely any virus in heart and lung (Fig 4D), which was consistent with previous findings showing that EV71 has a strong tropism to cause severe lesions in muscle tissues, but not in heart and lung tissues. In addition, we found that EV71 infection upregulated the expression of IFN-α in spinal muscles, as well as IFN-β in spinal muscles and hind leg muscles (Fig 4E), which are the inducers of TRIM21 and SAMHD1 genes. Therefore, we observed that the expression of TRIM21 and SAMHD1 was significantly increased in both spinal muscles and hind leg muscles (Fig 4F). These findings indicated that EV71 infection is positively associated with the expression of TRIM21 and negatively correlated with the expression of SAMHD1 in neonatal mice.

## The degradation of SAMHD1 induced by TRIM21 affects HIV replication

SAMHD1 was initially discovered as HIV-1 restriction factor [1,2]. To investigate the impact of TRIM21-mediated degradation of SAMHD1 on HIV-1 infection, we employed THP-1 cells, a human cell line derived from the peripheral blood of an acute monocytic leukemia patient that can be differentiated with phorbol esters (PMA). Sh-SAMHD1 or sh-TRIM21 THP-1 cells were constructed and differentiated by treatment with PMA for 24 h and then infected with VSV-G-pseudotyped reporter HIV. As expected, the depletion of SAMHD1 induced by SAMHD1 shRNA increased the sensitivity of THP-1 to HIV-1, while increased SAMHD1 induced by sh-TRIM21 blocked HIV-1 infection (Fig 5A–C). In monocyte-derived macrophage (MDM) cells, knockdown of TRIM21 also led to increased SAMHD1 (Fig 5D and E), which inhibited efficient HIV-1 infection (Fig 5F). However, in SAMHD1-negative Jurkat cells or in SAMHD1-positive KG-1 cells where SAMHD1 has no inhibitory effect on HIV-1 replication, also knockdown of TRIM21 had no influence on HIV-1 replication, further supporting the notion that the restriction on

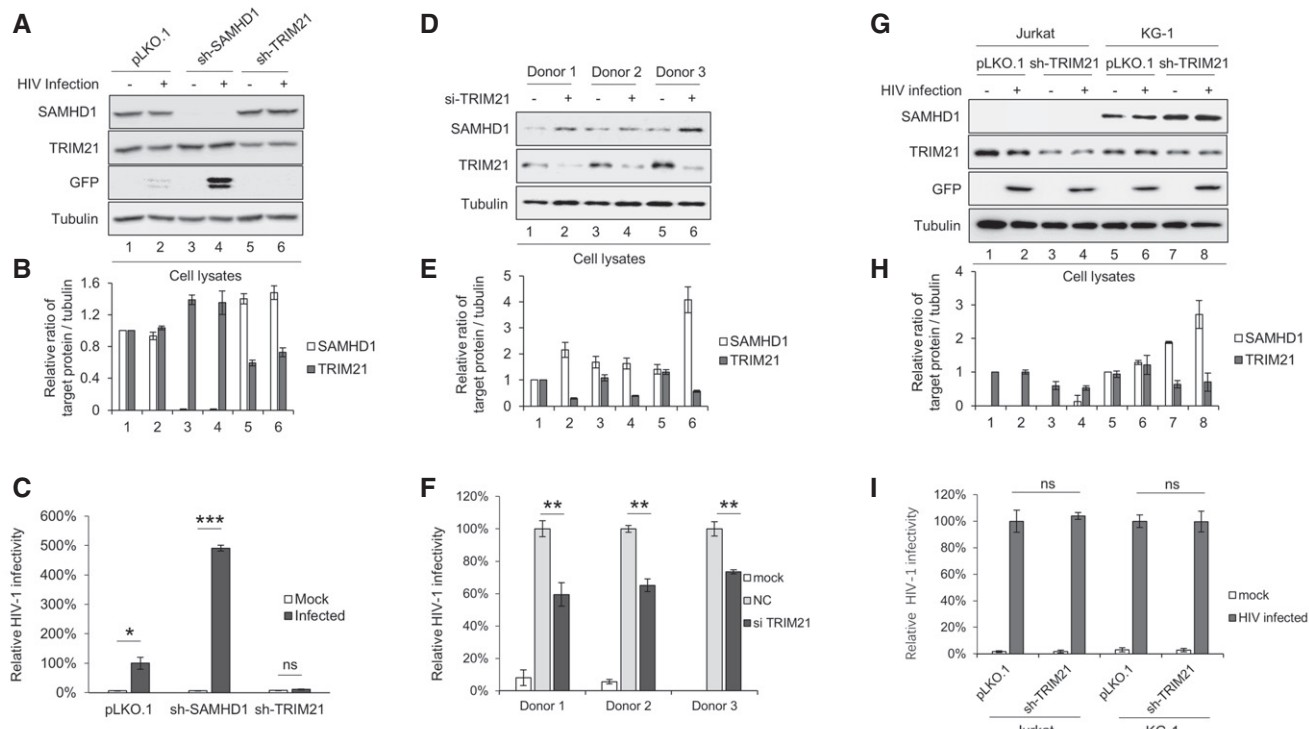

**Figure 5. The degradation of SAMHD1 by TRIM21 affects the anti-HIV-1 activity of SAMHD1.**

A–I (A) IB analysis of THP-1 cells treated with scrambled shRNA, TRIM21-specific shRNA, or SAMHD1-specific shRNA. Tubulin served as loading control. (B, E, H) Quantification of SAMHD1 and TRIM21 levels in whole cells in which the indicated genes were knocked down by shRNA. Data are normalized to tubulin and expressed as the fold change over cells treated with pLKO.1 without HIV infection ($n = 3$, mean ± SD). (C) sh-SAMHD1 and sh-TRIM21 THP-1 cells were treated with PMA for 24 h and infected with the VSV-G pseudotyped reporter HIV-1, and then detected 48 h later by flow cytometry analysis ($n = 3$, mean ± SD, *$P < 0.05$, **$P < 0.01$, ***$P < 0.001$, paired $t$-test). (D) IB analysis of MDM cells from three donors transfected with TRIM21 siRNA twice at days 1 and 3, respectively, after MDM differentiation. (F) MDM cells treated with TRIM21 siRNA were infected with VSV-G-pseudotyped HIV-1 reporter and then detected 48 h later by flow cytometry analysis ($n = 3$, mean ± SD, **$P < 0.01$, paired $t$-test). (G) IB analysis of Jurkat and KG-1 cells treated with scrambled shRNA or TRIM21-specific shRNA. (I) Jurkat and KG-1 cells treated with TRIM21 shRNA were infected with HIV-1 and then detected 48 h later by flow cytometry analysis ($n = 3$, mean ± SD, ns stands for no significance, paired $t$-test).

Source data are available online for this figure.

HIV-1 caused by TRIM21 overexpression or knockdown is SAMHD1-dependent (Fig 5G–I).

## The degradation of SAMHD1 induced by TRIM21 affects the inhibitory effect of SAMHD1 on IFN induction

Recent studies demonstrated that SAMHD1 promotes the degradation of nascent DNA at stalled replication forks and prevents IFN induction by limiting the release of single-stranded DNA, which activates the cGAS-STING pathway [61]. To evaluate the impact of TRIM21 regulation on the function of SAMHD1 in DNA repair, HEK293T cells transfected with TRIM21 were treated with the genotoxic agent hydroxyurea (HU), and the mRNA levels of IFN-α and IFN-β were detected. The reduction in SAMHD1 induced by TRIM21 led to increased levels of IFN-α and IFN-β mRNA (Fig EV4A–C), which gradually decreased with increasing SAMHD1 expression (Fig EV4D–F). However, a ligase-dead mutant TRIM21 C16A [62] had no effect on SAMHD1 expression and IFN production (Fig EV4D–F). Importantly, TRIM21-HA expression and SAMHD1 knockdown equally increased IFN-α and IFN-β signaling (Fig EV4G–I). However, deleting SAMHD1 in the TRIM21-HA-expressing cells did not alter the mRNA levels of IFN-α and IFN-β, indicating that IFN induction by TRIM21 happens in a SAMHD1-dependent manner (Fig EV4G–I).

## Identification of functional domains in TRIM21 and SAMHD1 required for their interaction

TRIM21 contains four domains: RING, B box, PRY, and SPRY. To determine the domain required for SAMHD1 interaction and degradation, we constructed truncated mutants and a single-amino-acid mutant of TRIM21 that loses the ability to bind zinc ions [62] (Fig 6A). The results showed that truncated TRIM21 and the C16A mutant lost the ability to degrade SAMHD1 (Fig 6B), while only truncated mutants without the PRY and SPRY domains could not interact with SAMHD1 (Fig 6C), indicating that PRY and SPRY are required for the interaction with SAMHD1 and also supporting the concept that the interaction is not enough to induce the degradation

of the target protein [63]. We next determined that SAMHD1 109–626 could be recognized and degraded by TRIM21 (Fig 6D and E), while SAMHD1 1–547 was resistant to TRIM21-mediated degradation, although it maintained the interaction with TRIM21 (Fig 6E and F). To substantiate the interaction, SAMHD1 109–626 and TRIM21-PRYSPRY recombinant proteins expressed in *Escherichia coli* also interacted with each other as illustrated in positive and reverse pull-down assays (Fig 6G). In order to verify the direct interaction, we also performed Fluorescence Resonance Energy Transfer (FRET) assays and found that after bleaching the signal from SAMHD1-YFP, ECFP fused with TRIM21 became brighter, but ECFP without TRIM21 remained unchanged (Fig 6H). Microscale thermophoresis assays (MST) also suggested that TRIM21-PRYSPRY directly interacts with SAMHD1 109–626 (Fig 6I).

To identify the particular TRIM21 recognition sites in human SAMHD1, we employed a panel of SAMHD1 orthologs from various species to examine the effect of TRIM21. Interestingly, *Canis* and *Gallus* SAMHD1 proteins were insensitive to TRIM21, while other SAMHD1 proteins were sensitive (Fig 7A). By full-length alignment analysis of various SAMHD1 proteins, we identified amino sites that are present only in the SAMHD1 proteins of *Canis* and *Gallus*, but not in other SAMHD1 proteins, and then made SAMHD1 mutants by site substitution (Fig 7B and C). By degradation and co-IP assays, we found that the amino acid sites G153 and G183 in SAMHD1 were required for TRIM21 interaction (Fig 7D and E).

## K622 in SAMHD1 is the ubiquitination site for TRIM21-mediated degradation

SAMHD1 1–547 maintained interaction with TRIM21, suggesting that residues 548–626 of SAMHD1 are important for degradation by TRIM21 (Fig 6F). Therefore, we speculated that some lysine sites located in SAMHD1 548–626 might be needed for the ubiquitination by TRIM21. To verify our assumption, we constructed arginine substitution mutants at each of the eight lysine sites in SAMHD1 548–626 (Fig 7F) and observed that all the mutants except K622R were sensitive to TRIM21 (Fig 7G). A co-IP assay confirmed that the

---

**Figure 6.  The interaction between TRIM21 and SAMHD1.**

A–C   TRIM21 interacts with SAMHD1 via PRY and SPRY domains. (A) Sketch map of TRIM21 WT and mutants. (B) The effect of TRIM21 on the degradation of SAMHD1. SAMHD1-flag was cotransfected with VR1012, TRIM21 WT, or the indicated mutant into HEK293T cells for 48 h, and the cells were subjected to IB with tubulin as loading control. (C) SAMHD1-flag was cotransfected with VR1012 or TRIM21 WT or the indicated mutant for 24 h, and the cells were then treated with 10 μM MG132 for 12 h before harvest and subjected to HA IP and IB.

D       Map of SAMHD1 WT and truncation mutants.

E       The effect of TRIM21 on SAMHD1 WT or its mutants. SAMHD1-HA or the indicated mutant were cotransfected with VR1012 or TRIM21-flag into HEK293T cells for 48 h, and the cells were subjected to IB with tubulin as a loading control.

F       SAMHD1-flag 1–547 was cotransfected with VR1012 or TRIM21-HA for 24 h, and the cells were then treated with 10 μM MG132 for 12 h before harvest and subjected to HA IP and IB.

G       SAMHD1 109–626 followed with a His tag or TRIM21-PRYSPRY followed with a GST tag was expressed in Rosetta (DE3), and pull-down assay was performed with Ni Sepharose (up) and GST Sepharose (down), respectively.

H       FRET analysis indicates interaction between YFP-SAMHD1 and CFP-TRIM21. A representative image of SAMHD1-YFP (yellow) and ECFP-TRIM21 (cyan)-expressing cells before and after photobleaching the acceptor fluorophore, YFP. The region chosen for photobleaching is marked (white open box), Bars, 10 μm. The quantization of fluorescence brightness was analyzed by ImageJ ($n = 3$, mean ± SD, ns stands for no significance, *$P < 0.05$, paired $t$-test).

I       The microscale thermophoresis curve for the interaction between SAMHD1 and TRIM21. Alexa Fluor® 647-labeled SAMHD1 at a concentration of 20 nM was incubated with twofold dilution series of unlabeled TRIM21 (3 μM to 9.16E-05 μM). The curve represents the signal recorded from three measurements. The normalized fluorescence thermophoretic signals were plotted against the concentration of TRIM21 with mean values ± standard deviation as well as a fitting to a 1:1 binding model (NanoTemper® Analysis software, F Norm = $F_{Hot}/F_{Cold}$). The $K_d$ of this interaction was determined to be 270 ± 92 nM. The data were a representative of three independent experiments using different dilution.

Source data are available online for this figure.

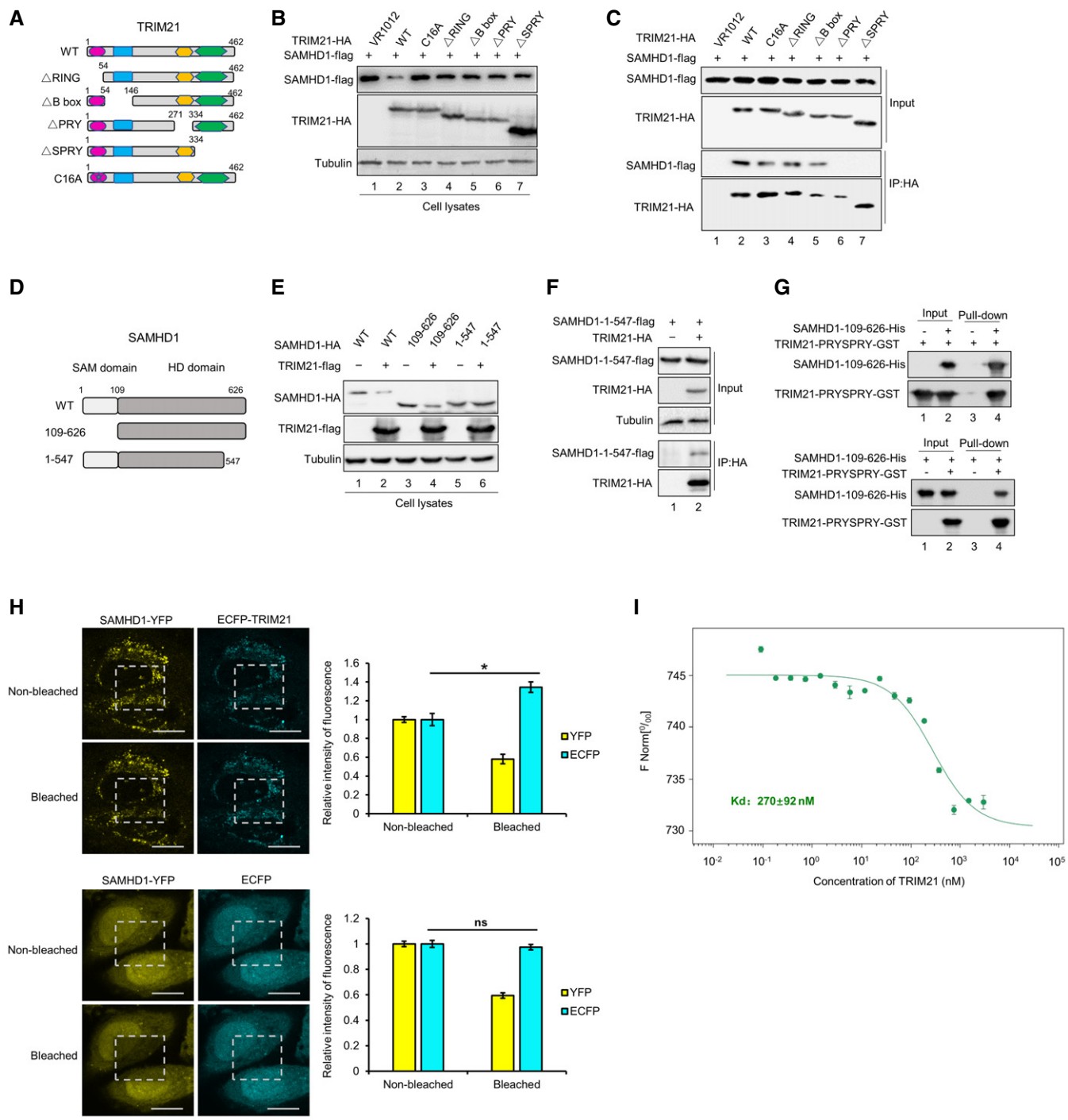

Figure 6.

ubiquitination of SAMHD1 induced by TRIM21 occurred via K48 linkage, while SAMHD1-K622R could not be ubiquitylated, in contrast to SAMHD1 WT (Fig 7H and I). The results demonstrated that TRIM21 delivers K48-ubiquitin to the K622 site of SAMHD1, and then, SAMHD1 is recognized and degraded by proteasomes. Finally, we observed that the K622R mutant of SAMHD1 which is resistant to TRIM21 degradation had a stronger ability to inhibit EV71 replication (Fig 7J).

## T592 phosphorylated SAMHD1 loses the ability to suppress EV71 replication

SAMHD1 not only inhibits HIV-1 reverse transcription by decreasing dNTP levels via its dNTPase activity [6,12], but also has RNase activity that contributes to HIV-1 restriction [16,17]. The major determinants required for dNTPase or RNase activities, as well as functional domains simulating different states of SAMHD1

oligomerization and phosphorylation have been characterized [17,64,65]. We overexpressed SAMHD1 mutants in sh-SAMHD1 HEK293T cells as indicated, and then cells were infected with EV71. We found that only T592D, which is the phosphorylation mimic mutant, could not repress EV71 replication in HEK293T cells (Fig EV5A), indicating that the restriction of EV71 by SAMHD1 is not associated with its dNTPase or RNase activity or even its oligomerization state, implying that an unknown mechanism of SAMHD1 inhibits EV71 replication. Based on this discovery, we further detected the levels of T592-phosphorylated SAMHD1 in both RD and HEK293T cells and observed less phosphorylated SAMHD1 in RD cells than in HEK293T cells (Fig EV5B), indicating that phosphorylated SAMHD1 is also sensitive to TRIM21. The overexpression of TRIM21 in HEK293T cells induced the degradation of endogenous phosphorylated SAMHD1, as well as ectopic phosphorylated and also unphosphorylated SAMHD1 (Fig EV5C and D). Moreover, TRIM21 knockdown had no effect on the expression of CDK2, phosphorylated CDK2, MCM2, as well as cyclin L2, which have been demonstrated to cause the degradation of SAMHD1 in MDM cells [25], indicating that TRIM21 directly regulates SAMHD1 expression by degrading SAMHD1, rather than altering cell cycle status or upregulating the expression of cyclin L2 (Fig EV5E and F). These data further support the conclusion that the regulation of SAMHD1 expression by TRIM21 is not associated with its phosphorylation state.

## Discussion

SAMHD1 has been the target of significant research and has been proven to possess multiple functions, including activity as a potent host restriction factor against retroviruses (such as HIV-1) and DNA viruses (like HSV-1) in noncycling cells [16,41,43,44]. However, whether SAMHD1 suppresses the replication of EV71, a major pathogen that causes HFMD, and whether some cellular factors regulate the expression and function of SAMHD1 have not yet been determined well. In this study, we demonstrate that SAMHD1 suppresses picornavirus EV71 replication by observing that EV71 shows different replication dynamics in HEK293T and RD cells (Fig 1). Importantly, we identified an essential regulator, the ubiquitin E3 ligase TRIM21, which was upregulated upon EV71 infection, as inducer of SAMHD1 degradation through the ubiquitin–proteasome pathway (Figs 2 and 3, and EV1). Further investigation discovered that EV71 infection stimulated IFN production which results in increased TRIM21 expression, suggesting a natural negative feedback of IFN response to limit SAMHD1 inhibition. In a neonatal mouse model, we further demonstrated that the expression levels of SAMHD1 and TRIM21 are negatively and positively correlated with tissue tropism upon EV71 infection, respectively (Fig 4). Moreover, the degradation of SAMHD1 induced by TRIM21, but not TRIM21 itself, affects the activity of SAMHD1 on EV71 restriction, HIV-1 restriction, and IFN production (Figs 5 and EV4). Our finding will be significant for understanding the interplay between host restrictive factor SAMHD1 and picornavirus as well as how viruses upregulate TRIM21 to regulate SAMHD1 level.

We also investigated the effect of SAMHD1 on other non-retro RNA viruses (RSV and VSV) and found that SAMHD1 had no effect on RSV replication, but inhibited VSV replication within 24 h, before being overcome by the virus proliferation (Fig EV2D–F). This result is partially consistent with the report by Choi et al [16] showing that SAMHD1 does not inhibit non-retro viruses such as Sendai virus, VSV, and reovirus.

Previous studies reported that TRIM21, as an interferon stimulating gene (ISG) and an E3 ligase, can be upregulated by pathogens and suppresses pathogens by interacting with the Fc of unique anti-pathogen antibodies, causing the ubiquitination and degradation of the pathogens through the proteasome pathway [35,38,54]. TRIM21 has also been implicated in the modulation of innate immunity, either as an innate immune enhancer or activator that positively regulates the antiviral response [37,39,56], or as an opposing factor that negatively regulates the innate immune response by inducing the degradation of DDX41 or facilitating the Nmi-IFI35 interaction which inhibits the innate antiviral response [36,40]. Here, we observed that EV71 infection upregulates TRIM21 expression by inducing type I IFN production (Fig EV3), whereas upregulated TRIM21 specifically degrades SAMHD1, which is associated with

**Figure 7. Binding sites and ubiquitination sites in SAMHD1 required for TRIM21 recognition.**

A  The effect of TRIM21 on SAMHD1 proteins from various species. HEK293T cells were transfected with VR1012 or TRIM21 and the indicated SAMHD1 expression vector and then subjected to IB analysis.

B  Identification of amino acids presented only in SAMHD1 of *Canis* and *Gallus* but not in other SAMHD1 proteins.

C  Construction of hSAMHD1 mutants with amino acid alterations.

D  The effect of TRIM21 on hSAMHD1 mutants. HEK293T cells were transfected with VR1012 or TRIM21 and SAMHD1 mutants for 48 h and then subjected to IB analysis.

E  SAMHD1-flag WT or the G153S or G183R mutant was cotransfected with VR1012 or TRIM21-HA for 24 h, and the cells were then treated with 10 μM MG132 for 12 h before harvest and subjected to HA IP and IB.

F  Sketch map of potential ubiquitination sites on SAMHD1.

G  SAMHD1-HA WT or the indicated mutants were cotransfected with VR1012 or TRIM21-flag into HEK293T cells for 48 h, and the cells were subjected to IB with tubulin as a loading control.

H  SAMHD1-HA WT and VR1012 or TRIM21 were cotransfected with K48-only and K63-only ubiquitin-flag into HEK293T cells for 24 h, and the cells were then treated with 10 μM MG132 for 12 h before harvest and subjected to HA IP and IB with tubulin as a loading control.

I  K48-only ubiquitin-flag and SAMHD1-HA WT or mutant K622R were cotransfected with VR1012 or TRIM21 into HEK293T cells as indicated. The cells were then treated with 10 μM MG132 for 12 h before harvest and subjected to HA IP and IB with tubulin as a loading control.

J  VR1012, SAMHD1-HA WT, or the K622R mutant was transfected into HEK293T-shSAMHD1 cells for 24 h. The cells were then infected with EV71 at a MOI of 0.1 and harvested at the indicated time points. SAMHD1-HA, TRIM21, and EV71-VP1 were detected by IB with tubulin as a loading control.

Data information: (A, D–E) Tubulin served as a loading control.
Source data are available online for this figure.

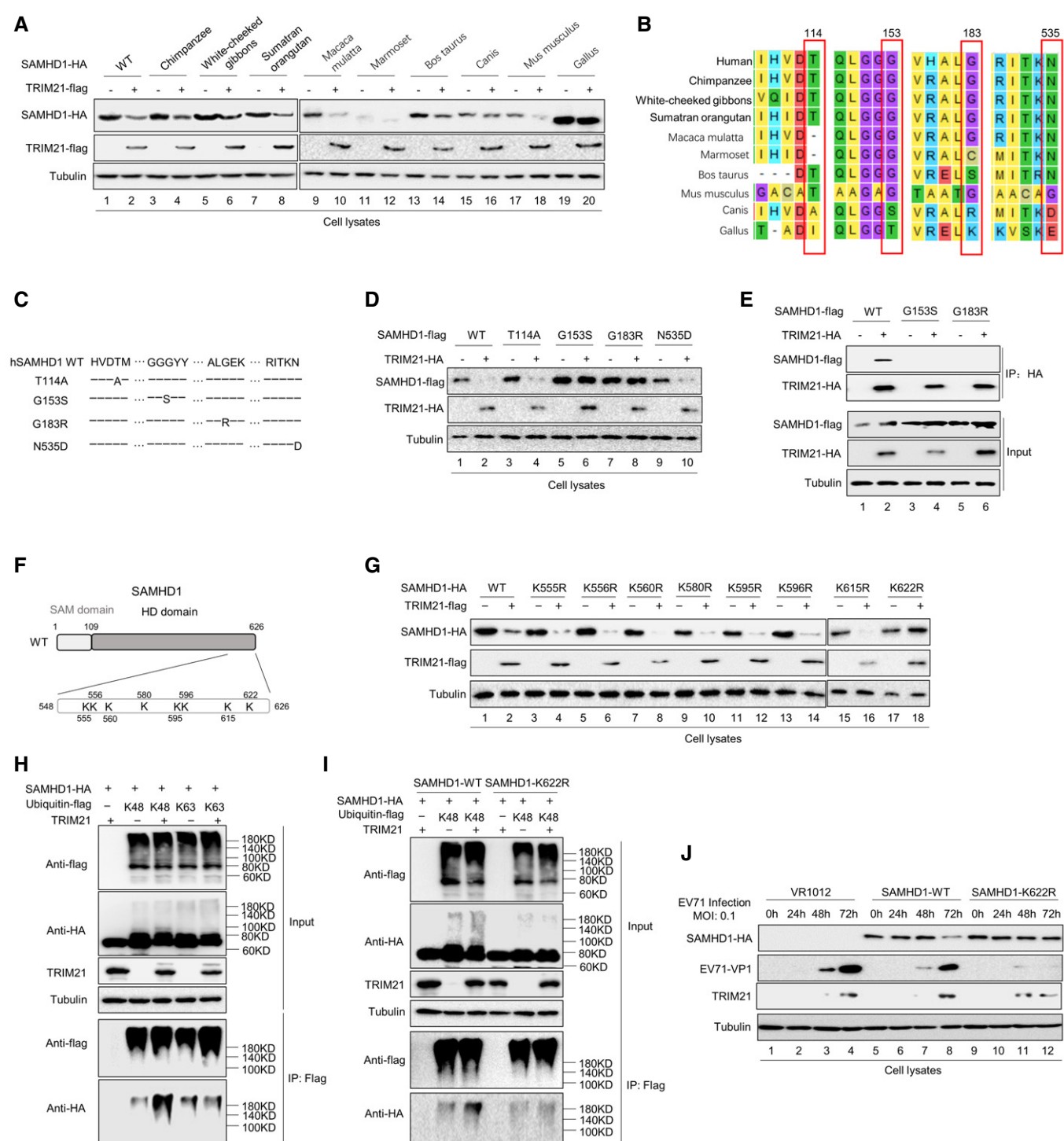

Figure 7.

the negative feedback regulation of the IFN responses. This phenomenon has been observed in a previous study in which TRIM21 was shown to ubiquitinate DDX41 for protein degradation [40]. Moreover, we found that the effect of TRIM21 on EV71 replication occurs in a SAMHD1-dependent manner (Fig 3F), which is inconsistent with the recent reports that TRIM21 restricts the replication of some viruses, such as coxsackievirus B3 (CVB3) binding to

MAVS and regulating IRF3-mediated type I IFN production [37]. Therefore, the function of TRIM21 varies among different viruses and plays diverse roles in viral replication due to its E3 ligase activity. Importantly, TRIM21 itself has no effect on IFN induction during DNA damage in the absence of SAMHD1 (Fig EV4).

In addition, we identified the functional domains required for their interaction in both TRIM21 and SAMHD1 (Figs 6–8). We found

that TRIM21 interacts with SAMHD1 via the PRY and SPRY domains, consistent with the observation that PRYSPRY domains are required for p62 binding [66]. SAMHD1 109–626 could be recognized and degraded by TRIM21, while the amino-terminal 1–547 of SAMHD1 maintained the ability to bind with TRIM21 but could not be degraded by TRIM21. Importantly, we confirmed the direct interaction between TRIM21 and SAMHD1 by pull-down assay on nickel or GST columns, FRET, and MST assays (Fig 6G–I), excluding nonspecific interactions between TRIM21 and the antibody. Further investigation revealed that TRIM21 ubiquitylates SAMHD1 at lysine 622 (K622) via K48 linkage and that G153 and G183 in SAMHD1 are required for TRIM21 recognition and degradation (Fig 7).

Mutants of SAMHD1 that lost their dNTPase or RNase activity maintained the ability to suppress EV71 replication, indicating that the depletion of dNTPs is not associated with the inhibition of EV71 by SAMHD1. In contrast, the T592D mutant, which is the SAMHD1 phosphorylation mimic, lost its restriction activity (Fig EV5A), which was also true for the inhibition of HIV [64,67]. Thus, how and where SAMHD1 intervenes in the life cycle of EV71 is needed to be investigated in the future. In addition, cell cycle transition in differentiated MDM cells with or without si-TRIM21 treatment was not affected as measured by the detection of MCM2 (Fig EV5).

The function and regulation of SAMHD1 in normal cells are an area of intense study. By depleting dNTPs, SAMHD1 acts as an inhibitor against retroviruses such as HIV, an effector of innate immunity and autoimmunity, and a regulator of cell cycle

progression [68,69]. The discovery that SAMHD1 is an effector of innate immunity was confirmed by mutations in the *SAMHD1* gene that cause Aicardi–Goutières syndrome (AGS) [10,70]. Recent studies have shown that SAMHD1 promotes DNA double-strand break repair and homologous recombination [22,71] and is also associated with HIV-1 latency [72]. Therefore, the role of TRIM21 as a SAMHD1 regulator may provide new insight into related research that will provide significant information about disease. Despite the recently discovered utility of TRIM21 as a tool to rapidly degrade endogenous proteins by utilizing antibody [34], our study suggests that the use of TRIM21 to degrade unique innate proteins might cause the degradation of other substrates of TRIM21 like SAMHD1.

In summary, we propose a mechanism in which EV71 infection upregulates TRIM21 expression, which specifically degrades the restriction factor SAMHD1 through the proteasome pathway and thereby overcomes the inhibitory effect of SAMHD1 on EV71 replication. In addition, the regulation of SAMHD1 by TRIM21 also affects the ability of SAMHD1 to block HIV-1 infection in THP-1 and MDM cells, as well as the function of SAMHD1 in IFN production during DNA damage (Fig 8). Therefore, these data suggest a possible mechanism employed by viruses to antagonize the inhibitory effect of a host defensive system, which broadens our understanding of the interplay between host and viruses, and provides information for the future development of innovative drug targets against virus infection.

# Materials and Methods

### Plasmid construction

mRNA of HEK293T and RD cells was extracted with TRIzol (Invitrogen, Carlsbad, CA, USA) and reverse-transcribed with oligo (dT) primers and M-MLV reverse transcriptase (Invitrogen) according to the manufacturer's instructions. The resulting cDNA was used for amplification of TRIM21; then, amplified fragment at C-terminal tagged with HA or flag was inserted into *SalI/BamHI* sites of VR1012 vector to generate the plasmid constructs TRIM21-HA and TRIM21-flag. Various truncated TRIM21 mutants or site mutations in TRIM21 were constructed by amplification and inserting into VR1012 by replacing TRIM21 WT or standard site-directed mutagenesis. Flag-Ubiquitin with K48 or K63 site mutation was amplified by PCR from myc-ubiquitin K48 or K63 plasmid which described previously [73] and constructed into *SalI/BamHI* sites of VR1012. SAMHD1-HA, SAMHD1-flag, SAMHD1-mCherry, Vpx-HA, and various site mutations in SAMHD1 were described previously [65,74,75]. Some truncated SAMHD1 mutants with HA tag or flag tag including SAMHD1 109–626, SAMHD1 120–626, and SAMHD1 1–547 were amplified from SAMHD1 plasmid and constructed into *SalI/BamHI* sites of VR1012. Various species SAMHD1s with HA tag were described previously [75]. SAMHD1 mutations Q548A, T592D, K555R, K556R, K560R, K580R, K595R, K596R, K615R, K622R, T114A, G153R, G183R, and N535D were obtained by using standard site-directed mutagenesis. SAMHD1 109–626 in pET28a-Plus vector with aHis6-tag at the N-terminus was described previously [65]. The DNAs encoding PRY&SPRY of human TRIM21 were generated by PCR with primers containing *EcoRI* and *XhoI* cleavage sites and cloned into vector pGEX-6P-1 with a GST tag at N-terminus. EV71

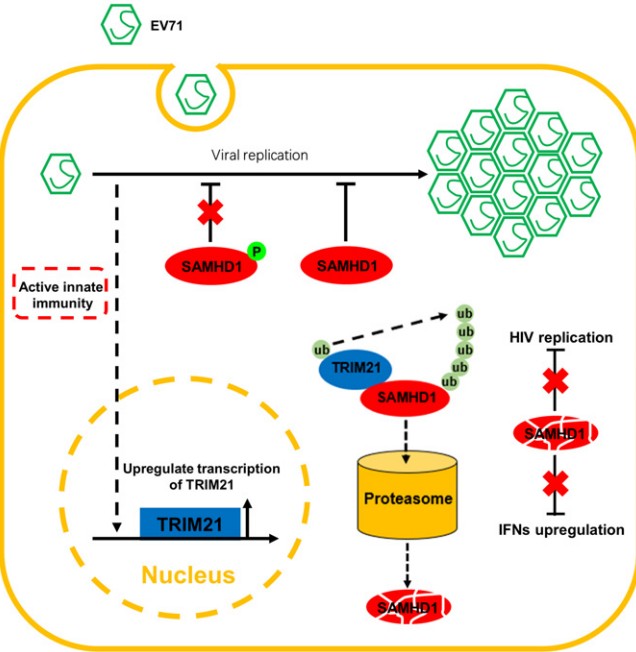

**Figure 8. Proposed mechanism of TRIM21-mediated degradation of SAMHD1.**

Host restriction factor SAMHD1 but not phosphorylated SAMHD1 restricts EV71 replication. However, EV71 infection overcomes the inhibitory of SAMHD1 by upregulating TRIM21, an E3 ubiquitin ligase, which induces the degradation of SAMHD1 via proteasome pathway. Moreover, the degradation of SAMHD1 induced by TRIM21 also affects its functions on HIV restriction and IFN production.

nonstructural viral proteins 2A-V5, 2B-HA, 2C-HA, 3A-HA, 3AB-HA, 3C-HA, and 3D-HA were described previously [76]. The TRIM25-flag plasmid was purchased from Addgene (Cambridge, MA, USA). All primers are shown in Appendix Table S2.

Retroviral vectors encoding SAMHD1-HA were constructed by amplification; then, amplified fragments were inserted into *XhoI/ BamHI* sites of pLVX-puro.

### shRNA construction

Three TRIM21-specific shRNAs with the following target sites were cloned in the lentiretroviral vector pLKO.1-puro (Addgene, Cambridge, MA, USA): shTRIM21-1, 5′-CCGGAATGCATCTCTCAGG TTGGGACTCGAGTCCCAACCTGAGAGATGCATTTTTTTG-3′ and 5′-AATTCAAAAAAATGCATCTCTCAGGTTGGGACTCGAGTCCCAACC TGAGAGATGCATT-3′; shTRIM21-2, 5′-CCGGAATCTAGGATTCACG CAGAGTCTCGAGACTCTGCGTGAATCCTAGATTTTTTTG-3′ and 5′-AATTCAAAAAAATCTAGGATTCACGCAGAGTCTCGAGACTCTGCG TGAATCCTAGATT-3′; shTRIM21-3, 5′-CCGGAAGAGTGGCTTCTG GACAATTCTCGAGAATTGTCCAGAAGCCACTCTTTTTTTG-3′ and 5′-AATTCAAAAAAAGAGTGGCTTCTGGACAATTCTCGAGAATTGT CCAGAAGCCACTCTT-3′. SAMHD1-specific shRNAs with the following target site were cloned in the lentiretroviral vector pLKO.1-puro. shSAMHD1: 5′-CCGGAATGTACACGCATGCTGAAGC CTCGAGGCTTCAGCATGCGTGTACATTTTTTTG-3′ and 5′-AATT CAAAAAAATGTACACGCATGCTGAAGCCTCGAGGCTTCAGCATGC GTGTACATT-3′. The SAMHD1 as well as TRIM21 siRNA sequences are same as shRNA and were purchased from RiboBio (Guangzhou, China). IFNAR-specific shRNAs with the following target site were cloned in the lentiretroviral vector pLKO.1-puro. shIFNAR: 5′-CCGGAAGAACTACAGCAGGACTTTGCTCGAGCAAAGTCCTGCTG TAGTTCTTTTTTG-3′ and 5′-AATTCAAAAAAGAACTACAG CAGGACTTTGCTCGAGCAAAGTCCTGCTGTAGTTCTT-3′.

### Stable silenced or overexpressing cell lines

For stable silenced cell lines, HEK293T cells were cotransfected with sh-SAMHD1-pLKO.1 or sh-TRIM21-pLKO.1 or pLKO.1 plus RRE, REV, and VSV-G expression vectors by using Lipofectamine 2000 (Invitrogen, Carlsbad, CA, USA). At 48 h after transfection, supernatants containing packaged lentivirus were harvested and used to infect HEK293T, RD, or THP1 cells for 48 h. Puromycin (3 μg/ml for HEK293T, 1 μg/ml for RD, and 1.5 μg/ml for THP1, Sigma, St. Louis, MO, USA) was then added into the culture to screen for stable cell lines. And for stable overexpressed cell lines, pLVX-SAMHD1-HA and pLVX vector were used as the same way.

### Knockout cell line construction

CRISPR-control (K010) lentiviral plasmids were obtained from Applied Biological Materials (Richmond, BC, Canada). CRISPR-TRIM21 was constructed by PCR with the primers 5′-AAAG GACGAAACACCTCATCTCAGAGCTAGATCGAGTTTTAGAGCTAGAA ATAG-3′ and 5′-AGGTGTTTCGTCCTTTCCACAAGAT-3′. Viruses contain gRNA and Cas9 were produced in HEK293T and then infected RD cells. After 48 h positive infection, RD cells were treated with 1 μg/ml puromycin. TRIM21 expression in survival cells was detected with Western blot.

### Cell culture and viruses

HEK293T (American Type Culture Collection [ATCC], Manassas, VA, USA, catalog no. CRL-11268), HepG2 (ATCC catalog no. HB-8065), Hela (ATCC catalog no. CRM-CCL-2), and human rhabdomyosarcoma RD (ATCC catalog no. CCL-136) cells were cultured as monolayers in Dulbecco's modified Eagle's medium (DMEM) (HyClone, Logan, UT, USA) supplemented with 10% heat-inactivated (56°C, 30 min) fetal calf serum (FCS, Gibco-BRL, Grand Island, NY, USA) and maintained at 37°C with 5% $CO_2$ in a humidified atmosphere. THP1 (ATCC catalog no. TIB-202) cells were maintained in RPMI 1640 medium with 10% FCS. Jurkat Clone E6-1 (ATCC, catalog no. TIB-152) and KG-1 (ATCC catalog no. CRL-8031) were maintained in RPMI 1640 medium with 10% FCS. EV71 CC063 strain was described in previous study [60]. The viral titer was determined in Vero cells according to the Reed–Muench formula. pNL4-3-△env-EGFP expression vector was a gift from R. Silicano. pVSV-eGFP expression vector was a gift from AH. Zheng (University of Chinese Academy of Sciences, Institute of Zoology), and respiratory syncytial virus (RSV) was a gift from CL. Jiang (Jilin University). All cell lines were tested for mycoplasma contamination.

### Monocyte isolation and differentiation

Peripheral blood mononuclear cells (PBMCs) were prepared from healthy donors (after informed consent was obtained), by density-gradient centrifugation (Lymphoprep, Axis-Shield, UK). Monocyte-derived macrophages (MDM) were prepared by adherence and discarding nonadherent cells 2 h later. Adherent cells were maintained in RPMI 1640 medium supplemented with 10% human AB serum and MCSF (15 ng/ml) for 6 days. For differentiation of THP1, cells were treated with 100 nM PMA for 24 h. Adherent cells were photographed using a microscope, and the number of adherent cells was counted.

### Transfection and infection

DNA transfections were carried out by Lipofectamine 2000 (Invitrogen) according to the manufacturer's instruction. TRIM21 siRNA transfections were carried out two times at days 1 and 3, respectively, by Lipofectamine RNAiMAX (Invitrogen) after MDM differentiation and then followed with HIV-1 infection at day 4.

After transfected with TRIM21 siRNA, PBMCs from donors were infected with EV71 CC063 strain or equal RPMI 1640 medium at 37°C for 1 h at indicated MOI, and then, PBMCs were washed with PBS and cultured in RPMI 1640 medium with 10% FCS.

For EV71 viral infection, briefly, cells were grown to 70% confluence in a 6-well plate, washed twice with phosphate-buffered saline (PBS), and incubated with EV71 CC063 strain at 37°C for 1 h at described MOI. During adsorption, the plate was gently agitated at 15-min intervals. Following adsorption, the virus-containing medium was replaced with fresh medium containing 2% FCS, followed by incubation at 37°C in 5% $CO_2$ for indicated time point. For HIV-1 viral infection, GFP-containing VSV-G pseudotyped HIV-1 was produced by transfecting pNL4-3ΔenvGFP plus VSV-G expression vector into HEK293T cells with Lipofectamine 2000 (Invitrogen). Forty-eight hours later, the supernatant was collected to infect THP-1 cells which were treated with PMA for 24 h. For VSV-GFP

and RSV infection, HEK293T-pLKO.1 and HEK293T-shSAMHD1 cell lines were incubated with VSV-GFP or RSV or equal DMEM at 37°C for 1 h at indicated MOI, and cells were washed with PBS and cultured in DMEM with 10% FCS.

## Immunoblotting (IB) and antibodies

Briefly, tissues from mice, transfected or infected HEK293T, RD, or Jurkat cells, were harvested and boiled in 1× loading buffer (0.08 M Tris, pH 6.8, with 2.0% SDS, 10% glycerol, 0.1 M dithiothreitol, and 0.2% bromophenol blue) followed by separation on a 12% polyacrylamide gel. Proteins were transferred onto a polyvinylidene fluoride (PVDF) membrane for Western blot analysis. The membranes were incubated with primary antibodies, followed by a corresponding horseradish peroxidase (HRP)-conjugated secondary antibody (Jackson ImmunoResearch, Suffolk, UK) diluted 1:10,000, respectively. Proteins incubated with HRP-conjugated secondary antibody were visualized using the ultrasensitive ECL Chemiluminescence Detection Kit (Proteintech, catalog no. B500024).

The following antibodies were used in this study: Polyclonal antibody (pAb) against EV71 was obtained from rabbits immunized with EV71 CC063 strain in our laboratory, anti-hemagglutinin (anti-HA) monoclonal antibody (mAb) (BioLegend, catalog no. 901513), anti-tubulin mAb (Abcam, catalog no. ab11323), anti-flag mAb (Sigma, catalog no. F1804), anti-BST2 pAb (OriGene, catalog no. TA310217), anti-SAMHD1 mAb (OriGene, catalog no. TA502024), anti-Phospho-SAMHD1 (Thr592) mAb (Prosci, Poway, catalog no. 8005), anti-TRIM21 pAb (Proteintech, catalog no. 12108-1-AP), anti-TRIM25 pAb (Abnova, Taiwan, catalog no. H00007706-B01P), anti-GFP pAb (Invitrogen, catalog no. A-21311), anti-His mAb (Proteintech, catalog no. 66005-1-Ig), anti-GST pAb (Proteintech, catalog no. 10000-0-AP), anti-RSV pAb (Millipore, catalog no. AB1128), anti-V5 mAb (Invitrogen, catalog no. R960-25), anti-CDK1 mAb (Abcam, catalog no. Ab32094), anti-CDK2 mAb (Abcam, catalog no. Ab32147), anti-MCM2 pAb (Proteintech, catalog no. 10513-1-AP), anti-Cyclin L2 pAb (Novus Biologicals, catalog no. NBP2-47497), Goat Anti-Rabbit IgG H&L (IRDye® 680RD) (Abcam, catalog no. Ab216777), and Goat anti-Mouse IgG H&L (IRDye® 800CW) (Abcam, catalog no. Ab216772). For confocal microscopy, Goat anti-Mouse IgG (H+L) Highly Cross Adsorbed Secondary Antibody, Alexa Fluor Plus 488 (Invitrogen, A32723), was used.

## RNA extraction and RT–qPCR

For RT–qPCR, viral or cell RNA was extracted from tissues of mice, HEK293T or RD cells transfected with SAMHD1 expression vector, or siRNA or cells infected with EV71 virus with TRIzol reagent (Invitrogen), diethyl pyrocarbonate (DEPC)-treated water, and RNase inhibitor (New England Biolabs, Ipswich, MA, USA). The cDNA was generated by a High-Capacity cDNA Reverse Transcription Kit (Applied Biosystems, Carlsbad, CA, USA) and oligo (dT) 18 primers according to the supplier's instructions. Reverse transcription was carried out in a 20 μl volume containing 1 μg of RNA extracted from samples and DNA digested by DNase (Promega, Madison, WI, USA). RT–qPCR was carried out on an Mx3005P instrument (Agilent Technologies, Stratagene, La Jolla, CA, USA)

with the RealMaster Mix (SYBR Green Kit, Takara, Shiga, Japan) and designed primers targeting the conserved region sequences of human A3A, A3B, A3C, A3D, A3F, A3G, A3H, SERINC5, BST2, SAMHD1, Mx2, GBP5, TRIM21, GAPDH, and EV71. The RT–qPCR assay was carried out in a 20 μl volume consisting of 9 μl of 2.5 × RealMaster Mix/20 × SYBR Green solution containing HotMaster Taq DNA Polymerase, 1 μl of 5 μmol/l of each oligonucleotide primer, and 2 μg of cDNA template. Amplification of the target fragment was carried out as follows: initial activation of HotMaster Taq DNA Polymerase at 95°C for 2 min, followed by 40 cycles of 95°C for 15 s, 57°C for 15 s, and 68°C for 20 s. All primers for qPCR were presented in Appendix Table S3.

## Pull-down assay

TRIM21-PRY&SPRY motif with GST tag and SAMHD1 109–626 with His tag were expressed in *E. coli* Transetta (DE3) Chemically Competent Cell (TransGen Biotech, Beijing, CHN), respectively, according to previous studies [31,65]. After treated with 0.5 mM IPTG in 16°C overnight, cells were harvested with buffer A (20 mM Tris–HCl and 50 mM NaCl) and clarified by ultrasonication. And then, lysates were centrifuged at 12,000 *g* for 30 min in 4°C. At the same time, the Ni Sepharose was incubated with 0.1 M NiSO$_4$ for 30 min at room temperature and washed with buffer A. After above operations, the supernatant of the lysates was harvested, mixed as designed, and incubated with Ni Sepharose or GST Sepharose (General Electric Company, Boston, MA, USA) for 4 h at 4°C. After 6-time wash with PBS for GST Sepharose pull-down assay or 50 mM imidazole solution for Ni Sepharose pull-down assay, Sepharoses were treated with 1× loading buffer and boiled. And then, target proteins were detected with immunoblot.

## Co-immunoprecipitation assay

The co-IP assays were performed as previously reported [73]. For TRIM21-HA immunoprecipitation, HEK293T cells transfected with SAMHD1-flag and TRIM21-HA or VR1012 were treated with MG132 (Sigma) for 12 h prior to harvest. The cells were then harvested and washed twice with cold PBS, followed by disruption with lysis buffer [PBS containing 1% Triton X-100 and complete protease inhibitor cocktail (Roche)] at 4°C for 1 h. Cell lysates were clarified by centrifugation at 10,000 × *g* for 30 min at 4°C. Anti-HA agarose beads (Roche) were mixed with the pre-cleared cell lysates and incubated at 4°C for 4 h on an end-over-end rocker. The reaction mixtures were then washed six times with cold wash buffer (20 mM Tris–HCl, pH 7.5, 100 mM NaCl, 0.1 mM EDTA, 0.05% Tween-20) and subsequently analyzed by immunoblotting. For TRIM21 or TRIM25 immunoprecipitation in RD-SAMHD1-HA cell line, TRIM21 antibody (Proteintech) or TRIM25 (Abnova) antibody and protein G agarose beads (Roche) were used.

## Mass spectrometry

RD stable overexpressed SAMHD1-HA cell line and control cell line were treated with MG132 (Sigma) or DMSO and performed co-IP assay with HA beads (Roche), and elutions were analyzed by mass spectrometry. Mass spectrum analysis was performed by ProtTech Biotechnology Company (Suzhou, China).

## Confocal microscopy

For localization of TRIM21 or TRIM25 or SAMHD1 alone, or colocalization of TRIM21 or TRIM25 with SAMHD1, Hela cells were transfected with TRIM21-flag or TRIM25-flag or SAMHD1-mcherry alone, or TRIM21-flag and SAMHD1-mcherry, or TRIM25-flag and SAMHD1-mcherry for 48 h. The cells were treated with 10 μM MG132 to avoid the degradation for 12 h prior to fixing, then fixed in 4% paraformaldehyde at room temperature for 15 min, washed with PBS, permeabilized in 0.1% Triton X-100 for 5 min, washed in PBS, blocked in 2% BSA for 1 h, and then incubated at room temperature for 2 h with mouse anti-flag antibody (Sigma) at 1:1,000. Following a wash, cells were incubated with Goat anti-Mouse IgG at room temperature for 1 h. After being washed with cold PBS, cells were analyzed by using a laser scanning confocal microscope (LSM810, Carl Zeiss, Oberkochen, Germany).

## Infection of newborn mice with EV71

All welfare and experimental procedures complied with the guidelines for the care and use of laboratory animals and were approved by the Ethics Review Committee of the First Hospital of Jilin University. EV71-infected newborn mice were established as previously described [60]. Briefly, 1-day-old specific-pathogen-free (SPF) ICR neonatal mice (Experimental Animal Center, Jilin University) were randomly divided into two groups (eight mice per group) and inoculated intracerebrally with EV71 CC063 viruses ($10^{6.5}$ $CCID_{50}$/ml) or MEM (10 μl/mouse). Then, the mice were anesthetized and subjected to analysis at 6 days post-infection. To minimize the effects of subjective bias, blinding of investigator was performed during group allocation and assessing results.

## Immunohistochemical analysis

As described in previous study [77], tissues and organs, including heart, lung, spinal muscle, and hind limb muscle, of the anesthetized mice were harvested and fixed in 3.7 % paraformaldehyde. The endogenous peroxidase activity of the tissues was inhibited by treatment with hydrogen peroxide (2.5 %). Amount of SAMHD1 in different tissues was detected by anti-SAMHD1 pAb (Proteintech, catalog no. 12586-1-AP) and a Streptavidin-Peroxidase Anti-Rabbit IgG Kit (Maixin), followed by color development with diaminobenzidine for detection of the antigen–antibody reaction at the same conditions.

## Fluorescence resonance energy transfer analysis

Hela cells in 6-well glass-bottom plates were transfected with SAMHD1-YFP (1 μg) and ECFP-TRIM21 (1 μg). Forty-eight hours later, cells were fixed in 4% paraformaldehyde at room temperature for 15 min and washed with PBS for three times. Fluorescent images of samples were then acquired with Olympus FV 3000 confocal imaging system.

Fluorescence intensity of photobleaching region was quantified with ImageJ software and represented with means ± SD from three measurements. Statistical significance was analyzed using Student's t-test (*P < 0.05).

## Microscale thermophoresis assay

Microscale thermophoresis experiments were performed on a NanoTemper® Monolith NT.115 (NanoTemper Technologies GmbH, Munich, Germany), capillary type: K002. SAMHD1 109–626 purified from E. coli was labeled with Alexa Fluor® 647 (Monolith NT™ Protein Labeling Kit RED-NHS) and kept at a concentration of 20 nM. Measurements were performed at 21.5°C using 40% MST power with laser on/off times of −1 s/0 s and 19 s/20 s, respectively. A twofold dilution series ranging from 0.1 nM to 3 μM of TRIM21 PRYSPRY purified from E. coli was prepared in PBS (pH 7.4), 0.05% (v/v) Tween. The experiments were repeated three times using different dilutions, and the data shown are a representative of three independent experiments using a single dilution series and was analyzed using the NanoTemper® analysis software (F Norm = $F_{Hot}/F_{Cold}$). The $K_d$ constant between SAMHD1 and TRIM21 was measured using the saturation binding curve at equilibrium.

## Ethics statement

PBMCs were from three healthy volunteer blood donors (two females and one male, 30–36 years old, mean 34). This study was approved by the Ethics Review Committee of the First Hospital of Jilin University and complied with the guidelines and principles of the WMA Declaration of Helsinki and the Department of Health and Human Services Belmont Report. Informed consent was signed by all research participants.

## Statistical analysis

The detailed statistical analysis has been described in figure legends. All data are shown as the mean ± standard deviations (SDs). Statistical comparisons between two groups were made using Student's t-test. Significant differences are indicated in the figures as follows: *$P \leq 0.05$, **$P \leq 0.01$, and ***$P \leq 0.001$. P-values of less than 0.05 are considered to represent a statistically significant difference. Ns stands for "no significance". Quantification of the bands in the immunoblot was performed using the ImageJ software.

# Data availability

The mass spectrometry data have been deposited to the PeptideAtlas (http://www.peptideatlas.org/PASS/PASS01460 for the sample from cells treated with DMSO and http://www.peptideatlas.org/PASS/PASS01465 for the sample from cells treated with MG132 with the dataset identifier PASS01460 and PASS01465.

Expanded View for this article is available online.

## Acknowledgements

We thank AH Zheng for VSV-GFP virus and CL Jiang for RSV virus. We thank L Wu for useful discussions and helpful suggestions. We thank the Protein Preparation and Identification Facility at the Technology Center for Protein Science of Tsinghua University for their assistance. We thank CY Dai for providing essential reagents. This work was supported in part by funding from the National Natural Science Foundation of China (No. 81672004, 31270202 and 81701987), the Jilin University Science and Technology Innovative

Research Team (JLUSTIRT, 2017TD-05), the Science and Technology Department of Jilin Province (20190101003JH), the Chinese Ministry of Science and Technology (2018ZX10302104-001-010), the Key Laboratory of Molecular Virology, Jilin Province (20102209), the Youth Foundation of the First Hospital of Jilin University (JDYY82017003), and the Graduate Innovation Fund of Jilin University.

## Author contributions

WZ and ZL conceived and designed the experiments, analyzed data, and wrote the manuscript. ZL, CH, HW, YL, and ZZ performed the experiments. XL and XS contributed to reagents, materials, and analysis tools. BZ, JY, and X-FY analyzed data.

## Conflict of interest

The authors declare that they have no conflict of interest.

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
