## [Review Process File · EMBO Reports]

TRIM21-mediated proteasomal degradation of SAMHD1 regulates its antiviral activity

Zhaolong Li, Chen Huan, Hong Wang, Yue Liu, Xin Liu, Xing Su, Jinghua Yu, Zhilei Zhao, Xiaofang Yu, Baisong Zheng and Wenyan Zhang

Review timeline:

Submission date:	4 December 2018
Editorial Decision:	14 December 2018
Resubmission	4 January 2019
Editorial Decision:	22 February 2019
Revision received:	8 May 2019
Editorial Decision:	5 June 2019
Resubmission	7 August 2019
Editorial Decision:	18 September 2019
Revision received:	9 October 2019
Accepted:	13 November 2019

Editor: Achim Breiling

Transaction Report:

1st Editorial Decision

14 December 2018

Thank you for the submission of your manuscript to EMBO reports. I read your work with interest and discussed it with my colleagues, and we decided to ask an expert advisor for an independent opinion. Taking into consideration his/her advice, I regret to say that we have decided that your manuscript is presently not well suited for our journal.

We appreciate that in your manuscript you report that SAMHD1 acts as a restriction factor for enteroviruses (EVs), and that the E3 ligase TRIM21 promotes EV replication by degrading SAMHD1. You also show that IFN production induced by EV infection upregulates TRIM21 expression, and that the degradation of SAMHD1 induced by TRIM21 affects the inhibitory effect of SAMHD1 on IFN induction. Further, you describe that TRIM21-dependent degradation of SAMHD1 also affects HIV replication in cultured cells. You also map the functional domains in TRIM21 and SAMHD1 required for their interaction, and identify K622 in SAMHD1 as the ubiquitination site for TRIM21-mediated degradation. Finally, you report that T592 phosphorylated SAMHD1 loses the ability to suppress EV71 replication, but that also this modified form is still degraded in a TRIM21-dependent manner.

We acknowledge that these data demonstrate that SAMHD1 suppresses EV replication, which extends the panel of viruses impacted by SAMHD1, that its levels are regulated by TRIM21-dependent ubiquitination. However, our advisor states that the data remain preliminary, as there are no *in vivo* analyses, even to show that TRIM21 is expressed in active innate immune cells or tissues either from mouse or human, and that virus infection *in vivo* is affected upon modulation of TRIM21 levels. The advisor feels that assessing TRIM21 distribution *in vivo* is essential for understanding the breadth of this regulatory process, and to proof physiological relevance. Considering these points, also as EMBO reports emphasises novel functional insight supported by

strong physiological experimental evidence, we do not think that the report presently provides the conceptual advance and broader impact we are looking for. We have therefore decided not to proceed with in-depth peer review.

However, in case you feel that you can address these concerns in a timely manner and could add in vivo data to further strengthen the manuscript as outlined above, we would have no objection to consider a new manuscript on the same topic in the near future. Please note that if you were to send a new manuscript this would be treated as a new submission rather than a revision and would be assessed afresh, also with respect to the literature and the novelty of your findings at the time of resubmission.

I am sorry to disappoint you this time, and I thank you once more for your interest in our journal.

Resubmission

4 January 2019

Enclosed please find our manuscript entitled "TRIM21-mediated proteasomal degradation of SAMHD1 regulates its antiviral activity" for your consideration for publication in *EMBO Reports*.

The restriction factor SAMHD1 has been reported to possess multiple functions including the restriction on retroviruses, DNA viruses, and retroelements, and fulfil important roles as effectors on innate and autoimmunity, a regulator of cell cycle progression and so on. However, whether cellular factors in host regulate the expression or functions of SAMHD1 has not been well characterized. Here, by investigating the effect of SAMHD1 on enterovirus 71 (EV71) which is one of the major pathogens caused Hand-Foot-Mouth Disease (HFMD) of children under 5 years old, we not only demonstrated for the first time SAMHD1 inhibits EV71 or coxsackievirus A16 (CVA16) replication, but also identified a SAMHD1-interacting protein TRIM21, an E3 ubiquitin ligase, which ubiquitylated and induced the degradation of SAMHD1 through a proteasome pathway. The mRNA and protein expression levels of SAMHD1 and TRIM21 were detected in different tissues of neonatal mouse and we found the opposite tendency between SAMHD1 expression and not only the susceptibility to EV71, but also the expression of TRIM21 in vivo. The novel mechanism evolved by EV71 through upregulating TRIM21 but not its non-structural proteins to antagonize host restriction factor SAMHD1, is a novel finding which will be significant for broadening our insights on the interplay between viruses and host restriction factors. Moreover, to investigate the essential role of TRIM21 in SAMHD1's other biochemistry and biology functions will provide the stratagem to regulate SAMHD1 and help our understanding for related disease.

Our results represent one of the most interesting developments in the researches on host defensive factors and picornaviruses. Our results merging several research fields including virology, innate immunity, and cell biology, should be of broad interest for investigators from diverse fields including virologists, cell biologists, structural biologists, and drug development researchers. None of the material presented in this manuscript has been published or is under consideration for publication elsewhere.

2nd Editorial Decision

22 February 2019

Thank you for the submission of your research manuscript to EMBO reports. We have now received reports from the three referees that were asked to evaluate your study, which can be found at the end of this email.

As you will see, all referees think that your manuscript is of interest to the readership of EMBO reports, but requires significant revision to be considered for publication. All referees have several concerns and suggestions to improve the manuscript. As the reports are below, I will not detail them here. Nevertheless, I think that the major point of referee #1, concerning the direct interaction of TRIM21 and SAMHD1, needs substantial further experimental proof (see also point 4 of referee #2). Further, in particular the major points of referee #3 need to be addressed experimentally.

Given the constructive referee comments, I would like to invite you to revise your manuscript with

the understanding that all referee concerns must be addressed in the revised manuscript and in a detailed point-by-point response. Acceptance of your manuscript will depend on a positive outcome of a second round of review. It is EMBO reports policy to allow a single round of revision only and acceptance or rejection of the manuscript will therefore depend on the completeness of your responses included in the next, final version of the manuscript.

Revised manuscripts should be submitted within three months of a request for revision; they will otherwise be treated as new submissions. Please contact us if a 3-months time frame is not sufficient for the revisions so that we can discuss the revisions further.

Supplementary/additional data: The Expanded View format, which will be displayed in the main HTML of the paper in a collapsible format, has replaced the Supplementary information. You can submit up to 5 images as Expanded View. Please follow the nomenclature Figure EV1, Figure EV2 etc. The figure legend for these should be included in the main manuscript document file in a section called Expanded View Figure Legends after the main Figure Legends section. Additional Supplementary material should be supplied as a single pdf labeled Appendix. The Appendix includes a table of content on the first page, all figures and their legends. Please follow the nomenclature Appendix Figure Sx throughout the text and also label the figures according to this nomenclature.

For more details please refer to our guide to authors:
<http://embor.embopress.org/authorguide#manuscriptpreparation>

Important: All materials and methods should be included in the main manuscript file.

See also our guide for figure preparation:
http://www.embopress.org/sites/default/files/EMBOPress_Figure_Guidelines_061115.pdf

Regarding data quantification and statistics, can you please specify, where applicable, the number "n" for how many independent experiments (biological replicates) were performed, the bars and error bars (e.g. SEM, SD) and the test used to calculate p-values in the respective figure legends. Please provide statistical testing where applicable. See:
<http://embor.embopress.org/authorguide#statisticalanalysis>

We now strongly encourage the publication of original source data with the aim of making primary data more accessible and transparent to the reader. The source data will be published in a separate source data file online along with the accepted manuscript and will be linked to the relevant figure. If you would like to use this opportunity, please submit the source data (for example scans of entire gels or blots, data points of graphs in an excel sheet, additional images, etc.) of your key experiments together with the revised manuscript. Please include size markers for scans of entire gels, label the scans with figure and panel number, and send one PDF file per figure.

- a complete author checklist, which you can download from our author guidelines (<http://embor.embopress.org/authorguide#revision>). Please insert page numbers in the checklist to indicate where the requested information can be found.
- a letter detailing your responses to the referee comments in Word format (.doc)
- a Microsoft Word file (.doc) of the revised manuscript text
- editable TIFF or EPS-formatted single figure files in high resolution (for main figures and EV figures)

Please also note that we now mandate that all corresponding authors list an ORCID digital identifier that is linked to their EMBO reports account!

I look forward to seeing a revised version of your manuscript when it is ready. Please let me know if you have questions or comments regarding the revision.

REFeree REPORTS

Referee #1:

The manuscript by Li et al. describes work seeking to establish a connection between SAMHD1 and the cytosolic Fc receptor TRIM21. Specifically, the authors propose that TRIM21 negatively regulates SAMHD1 and is induced to do so by EV71. This is a somewhat surprising claim, given that TRIM21 is an antiviral ISG that is upregulated by interferon induced by the host as a response to infection (something that was already well-established before the present manuscript). It doesn't make much sense for the host to up regulate an ISG to promote infection. SAMHD1 is regulated by the cell cycle, as dNTPs are not required in the cytosol except during cell division. To what purpose would TRIM21 have evolved to interfere with this critical cell cycle regulation? The findings here don't fit with any of the published literature on SAMHD1 or TRIM21 and no rationale is provided by the authors to address this.

I also have serious concerns with the data in this manuscript. A very large warning sign that the interaction between TRIM21 and SAMHD1 is spurious is the fact that it was identified by co-IP using antibodies against SAMHD1. TRIM21 binds antibodies with sub-nanomolar affinity, therefore of course it will IP. Unfortunately this is not the first time this mistake has been made and in every former case the response has been that it must be real because the reverse IP confirms it. As this approach has repeatedly failed to distinguish real from artifactual interaction it absolutely cannot be relied upon. To demonstrate direct interaction between TRIM21 and SAMHD1 there must be rigorous interaction evidence obtained using the recombinant proteins and the affinity quantified (eg by ITC or SPR etc). As both proteins are easy to express, this is straightforward.

Minor points

1. I don't see a scrambled shRNA control to compare TRIM21 depletion to.
2. Fig 4. Authors used comparative ($\Delta\Delta CT$) Q-PCR method to examine mRNA level of SAMHD1 and TRIM21 in heart, lungs, spinal muscles and hind leg muscles. The expression of SAMHD1 in heart was set as 100%, then they somehow compared TRIM21 level to SAMHD1- it's wrong, as these are 2 different target genes. It is only possible to compare expression level of the same target in different tissues.
3. Though a fundamental part of the manuscript, the data that TRIM21 regulates SAMHD1 expression is particularly weak. An incomplete shRNA knockdown to show a minimal effect is not convincing. I don't know why CRISPR KO hasn't been used instead of shRNA.
4. Overexpression of TRIM21 is known to cause artifactual aggregation in the cell and this is a particular problem when it is expressed from a strong viral promoter.
5. The co-localisation data seems strange - especially the localization of T21 at the nuclear membrane. TRIM21 expression should be diffuse and cytosolic.

Referee #2:

In this work, Zhaolong Li et al, describe how the E3 ubiquitin ligase regulates the activity of SAMHD1 upon Enterovirus 71 infection in two different cell lines. Zhaolong Li et al., claim that SAMHD1 is a restriction factor for EV71. TRIM21 mediates the degradation of SAMHD1 acting as a proviral protein. The results are intriguing, but some sections of the work are difficult to follow and understand since this is a long manuscript with an extensive number of figures, and there is a lack of experimental details, controls and some caveats that dampen their conclusions.

1. In general, since the authors support most of their results in western blot experiments, I would suggest adding a quantification of protein bands of the gel (i.e. by Image J), since in some figures there is no a clear expression of the proteins. As an example, figure 5 or figure 1D for HepG2.
2. Material and methods should be better explained. There is no statistics analysis section anywhere. Figure legends are not detailed explained and there are some errors. For example, in figure suppl 5C and D, there is no tubulin control although it is written in the legend. Or in figure 5E, where they use VR1012 as a control. Why do the authors use pLKO.1?
3. The authors use in this work two main cell lines, RD and HEK293T cells. It is very confusing when and why they use one cell line or the other. It should be explained better.

4. Authors could perform some of the experiments under endogenous levels of expression of the proteins. Pull down experiments in figure 2 could be done with endogenous TRIM21, at least in RD cells where they claim a major expression of the protein. The same for the microscopic images in 2I (in addition, authors should add another irrelevant trim, as a control).
5. Published reports claim that SAMHD1 does not inhibit the replication of non-retro RNA viruses (i.e. Choi J et al 2015. DOI: 10.1186/s12977-015-0174-4). Experiments such as adding SAMHD1 and then monitoring the diminution of viral RNA/protein levels could help to strengthen their conclusions.
6. Some publications suggest TRIM21 as an activator of innate immune responses and antiviral factor through its interaction (Liu H et al. Front Immunol.2018 DOI: 10.3389/fimmu.2018.02479) with RLR signaling factors, such as RIG-I and MAVS (Xue B, et al. JVirol 2018. DOI: 10.1128/JVI.00321-18). Could the authors exclude this?
7. It is known that proteases in picornavirus can play an important role inhibiting key steps in the activation of innate immune responses. Can the authors exclude an effect of some viral protein in the context of infection in their work? (Wang B et al, Plos Pathog 2013. DOI: 10.1371/journal.ppat.1003231). Supplemental Figure3E is not conclusive.
8. There is the same amount of mRNA levels in both cell lines in figure 1C for SAMHD1 and other restriction factors, with and without infection. How authors explain this?
9. In figure 1E, authors should include knock down of SAMHD1 in RD cells. Are the experiments performed in 293T, or RD cells?
10. I would suggest using another TRIM family protein as control, in the experiments done in figure 3 to show that the results are specific of TRIM21.
11. Why there is a difference in induction of IFNB in figure 4?

Referee #3:

In this manuscript, the authors report that cellular SAMHD1 can restrict enterovirus 71 infection. Follow-up mass spec and co-IP experiments indicated that SAMHD1 interacts with TRIM21, indicating a possible functional connection between SAMHD1 and TRIM21. Consistent with the hypothesis that TRIM21 targets SAMHD1 for proteasomal degradation, cell lines and neonatal mouse tissues with lower SAMHD1 expression were more susceptible to EV71 infection.

Co-IP approaches with TRIM21 and SAMHD1 domain mutants, indicated that these protein interact through the PRY-SPRY domain of TRIM21 and the N-terminal 1-547 aa of SAMHD1. Despite being sufficient for interaction with TRIM21, the 1-547 aa region of SAMHD1 was not degraded by TRIM21, consistent with the extreme SAMHD1 C-terminus being ubiquitinated by TRIM21. Cell-based assays with K-to-R point mutants indicated that K622 in the SAMHD1 C-terminus is ubiquitinated by TRIM21. Taken together, the authors conclude that EV71 induces TRIM21 as an antagonistic mechanism to decrease cellular SAMHD1 levels, thereby favoring its infective cycle.

1. Does this manuscript report a single key finding?
 Yes; TRIM21 degrades the SAMHD1 viral restriction factor, thereby favoring EV71 infection.
2. Is the reported work of significance, or does it describe a confirmatory finding or one that has already been documented using other methods or in other organisms etc?
 Yes; while it has been reported that SAMHD1 can restrict retroviruses, this work shows that unrelated RNA viruses are also restricted by SAMHD1, and that TRIM21 mechanistically controls SAMHD1 abundance.
3. Is it of general interest to the molecular biology community?
 Yes; SAMHD1 is an interesting factor because of its AGS-disease association, as well as retroviral restriction. How SAMHD1 is regulated, and that it targets other viral species is in this reviewer's opinion interesting for a broad audience.
4. Is the single major finding robustly documented using independent lines of experimental evidence?
 For a large part the presented data is compelling, and as a whole supports the main conclusion. However, several key experiments are either not performed or cannot be properly interpreted in their

current form.

Main comments for authors:

A) The effect in EV71 infection is measured through viral RNA or VP1 protein levels. To understand if SAMHD1 and/or TRIM21 levels influence infection in a biological meaningful way, at least one key experiment should include a measurement of infectious virus.

B) The presented data reasonably supports that TRIM21 and SAMHD1 individually influence EV71 infection, and that they interact. However, in this reviewer's opinion, genetic epistasis evidence (e.g. Fig. 3H, 6-G-I) is not very strong.

C) Given the plethora of cellular roles described for both TRIM21 and SAMHD1, it is unclear whether their manipulation affects the general fitness of the cell to support viral replication, or affects specifically EV71.

D) The conclusion that EV71 has developed IFN-dependent TRIM21 induction as an antagonism strategy to degrade SAMHD1 seems far-fetched in this reviewer's opinion. The general thought is that type I IFNs are the key antiviral response for practically all viruses, and that virtually all viruses induce type I IFNs during their life cycle, yet have developed viral antagonists to limit this. In this reviewer's opinion, it seems much more likely that TRIM21 would be a cellular negative feedback mechanism to limit SAMHD1 levels after type I IFN exposure. Since EV71 (as most viruses) induces IFN β , it also induces this response.

E) In this reviewer's opinion, cannot be interpreted for the following reasons:

- There are no MW markers, making it unclear whether the IPed SAMHD1 bands indeed exclusively are of SAMHD1+at least 8.5 kDa (for Ub).

- Ubiquitination can only be shown if one pulls on Ub and blots for SAMHD1, or inverse. Here the IP is for SAMHD1 and the WB as well.

- The Ub and TRIM21 constructs have both the same tag, there should be a dark smear of ubiquitinated cellular proteins in the TRIM21 input (which is not there).

- The "ubiquitin-flag" construct is not described in the manuscript, but the name suggests that it may have a C-terminal tag (which would be unable to conjugate, and could explain the lack of protein smear as pointed out in the point above).

In order to fully substantiate the claims made in the manuscript, this reviewer's opinion the following experiments should be included (letters correspond to main comments above):

A+B): Epistasis experiment in which Trim21 and Samhd1 are individually depleted, as well as in combination, followed by EV71 infection, with sampling of produced virus in a growth curve (read out by plaque assay or TCID50). Samhd1 knock-down seems efficient, but for Trim21, a CRISPR knock-out line (or MEFs from Trim21^{-/-} mice) would be suggested.

C): Determine the effect of Samhd1 and Trim21 knock-down and over-expression on multi-cycle infection with an unrelated virus (e.g. VSV).

D): To get an indication if TRIM21 upregulation is a viral antagonism strategy, the authors could determine the effect of ablating the IFN-response (Ifnar^{-/-}) on viral titers, compared to Samhd1^{-/-} and restoration of Samhd1 in the Ifnar^{-/-} background. It stands to reason that the strongest effector has provided the strongest evolutionary pressure to be antagonized.

E): A His-Ub denaturing pull-down (urea; with Ni-NTA), followed by a blot for SAMHD1 should be performed. Blots of the input and PD for His-Ub and SamHD1-HA should be shown. MW size markers should be carefully indicated (as well as in the rest of the figures); there should be no unmodified SAMHD1 in the pull-down.

E2): Given the moderately effective knock-down of Trim21, and mild effects on SAMHD1

expression, a Trim21 CRISPR knock-out line in which SAMHD1 stability is measured (compared to non-targeted cells) in the absence and presence of MG132, should be included.

Other comments:

- 1) Line 141: "nucleUS to nuclear membrane"?
- 2) Line 200: "By" Delete word?
- 3) Lines 230-235: To the best of my knowledge HEK-293Ts do not have a functional cGAS-STING pathway.
- 4) Alignment issue in figure panel 6A.
- 5) The interpretation of Fig. 6G-I is unclear. My interpretation is that TRIM21-HA expression and Samhd1 knock-down each equally increase the reporter. Yet, deleting Samhd1 in the TRIM21-HA expressing cells does not alter the reporter induction, suggesting that that the induction by TRIM21-HA is INdependent of Samhd1.
- 6) Fig 7C: Unclear why there is no decrease in SAMHD1 when TRIM21 is expressed.
- 7) Fig 7F: The labeling for the top plasmid is missing.
- 8) The neonatal mouse figure seems a bit out of place at the current position, since it essentially demonstrates that there is altered Samhd1/Trim21 expression at the sites of infection. As there is not genetics, and contribution to causality, it may better fit earlier in the manuscript (e.g. with Fig. 1) to make a case for correlation between the two factors and infection susceptibility.
- 9) Line 310/311: Without cell-free in vitro reconstitution it cannot be claimed that TRIM21 directly targets SAMHD1 for degradation.
- 10) Line 378-381: Not entirely clear what is meant here. TRIM-Away is based on intra-cellular antibody-directed targeting of TRIIM21 to the target of the antibody. Since this does not alter TRIM21 levels itself, it is difficult to see why this would cause off-targets to endogenous TRIM21 targets such as SAMHD1.

1st Revision - authors' response

8 May 2019

Referee #1:

The manuscript by Li et al. describes work seeking to establish a connection between SAMHD1 and the cytosolic Fc receptor TRIM21. Specifically, the authors propose that TRIM21 negatively regulates SAMHD1 and is induced to do so by EV71. This is a somewhat surprising claim, given that TRIM21 is an antiviral ISG that is upregulated by interferon induced by the host as a response to infection (something that was already well-established before the present manuscript). It doesn't make much sense for the host to up regulate an ISG to promote infection. SAMHD1 is regulated by the cell cycle, as dNTPs are not required in the cytosol except during cell division. To what purpose would TRIM21 have evolved to interfere with this critical cell cycle regulation? The findings here don't fit with any of the published literature on SAMHD1 or TRIM21 and no rationale is provided by the authors to address this.

Response: We thank the reviewer for the question. Indeed, previous studies had showed TRIM21 mediates dual effector and sensor functions by facilitating simultaneous proteasomal degradation of virions and innate immune signaling [1-3]. However, as an E3 ligase, TRIM21 has also been reported to negatively regulate the innate immune response to dsDNA by degrading DDX1, a sensor of dsDNA [4]. TRIM21 also contributed to the negative regulatory function of the Nmi-IFI35 complex on innate antiviral signaling [5]. Thus, the role of TRIM21 in viral infection is more complicated due to its E3 ligase activity. TRIM21 and SAMHD1 both are IFN-inducible, here, we demonstrate that EV71 infection induced IFN production which induces TRIM21 to antagonize the restrictive effect of SAMHD1 on EV71 replication, which had never been reported before, further indicating that our innovative discovery is significance for understanding the interplay between host and virus. We discussed it in discussion part.

I also have serious concerns with the data in this manuscript. A very large warning sign that the interaction between TRIM21 and SAMHD1 is spurious is the fact that it was identified by co-IP using antibodies against SAMHD1. TRIM21 binds antibodies with sub-nanomolar affinity, therefore of course it will IP. Unfortunately, this is not the first time this mistake has been made and in every former case the response has been that it must be real because the reverse IP confirms it. As this approach has repeatedly failed to distinguish real from artifactual interaction it absolutely cannot be

relied upon. To demonstrate direct interaction between TRIM21 and SAMHD1 there must be rigorous interaction evidence obtained using the recombinant proteins and the affinity quantified (eg by ITC or SPR etc). As both proteins are easy to express, this is straightforward.

Response: We thank the reviewer for the great suggestion. We expressed the two proteins with His and GST tag respectively, in *E.coli* system, then employed positive and reverse pull down assays with Nickel and GST column and confirmed the direct interaction between TRIM21 and SAMHD1 (Fig.7G).

Minor points

1. I don't see a scrambled shRNA control to compare TRIM21 depletion to.

Response: pLKO.1 expression vector with a scrambled shRNA was used as a negative control to compare with TRIM21 depletion in this manuscript.

2. Fig 4. Authors used comparative ($\Delta\Delta$ CT) Q-PCR method to examine mRNA level of SAMHD1 and TRIM21 in heart, lungs, spinal muscles and hind leg muscles. The expression of SAMHD1 in heart was set as 100%, then they somehow compared TRIM21 level to SAMHD1- it's wrong, as these are 2 different target genes. It is only possible to compare expression level of the same target in different tissues.

Response: We thank the reviewer for the great suggestion. We corrected the mistake and set the expression of TRIM21 in heart as 100%.

3. Though a fundamental part of the manuscript, the data that TRIM21 regulates SAMHD1 expression is particularly weak. An incomplete shRNA knockdown to show a minimal effect is not convincing. I don't know why CRISPR KO hasn't been used instead of shRNA.

Response: We thank the reviewer for the great suggestion. We constructed TRIM21 CRISPR KO to further confirm the data (Fig.3).

4. Overexpression of TRIM21 is known to cause artifactual aggregation in the cell and this is a particular problem when it is expressed from a strong viral promoter.

Response: We thank the reviewer for the suggestion. We actually can't exclude the possibility of the artificial aggregation due to TRIM21 overexpression. However, we also provided the data that knockdown or knockout of TRIM21 increased SAMHD1 expression and decreased EV71 replication, as well as overexpression of TRIM25 had no effect on SAMHD1 expression. So, we concluded that the effect of TRIM21 in this project is not artificial effect.

5. The co-localisation data seems strange - especially the localization of T21 at the nuclear membrane. TRIM21 expression should be diffuse and cytosolic.

Response: TRIM21 is a cytosolic protein. Here, we speculate that the localization of TRIM21 might to be altered upon EV71 infection, which is consistent with the report that the localization of some of TRIMs was found to be altered during viral infection [3].

Referee #2:

In this work, Zhaolong Li et al, describe how the E3 ubiquitin ligase regulates the activity of SAMHD1 upon Enterovirus 71 infection in two different cell lines. Zhaolong Li et al., claim that SAMHD1 is a restriction factor for EV71. TRIM21 mediates the degradation of SAMHD1 acting as a proviral protein. The results are intriguing, but some sections of the work are difficult to follow and understand since this is a long manuscript with an extensive number of figures, and there is a lack of experimental details, controls and some caveats that dampen their conclusions.

Response: We thank the reviewer for the great suggestion. We tried to make the manuscript more clear, like adding the experimental details, showing controls et al.

1. In general, since the authors support most of their results in western blot experiments, I would suggest adding a quantification of protein bands of the gel (i.e. by Image J), since in some figures there is no a clear expression of the proteins. As an example, figure 5 or figure 1D for HepG2.

Response: According to the reviewer's suggestion, we quantified the protein bands of the gel by Image J software.

2. Material and methods should be better explained. There is no statistics analysis section anywhere. Figure legends are not detailed explained and there are some errors. For example, in figure suppl 5C and D, there is no tubulin control although it is written in the legend. Or in figure 5E, where they use VR1012 as a control. Why do the authors use pLKO.1?

Response: We thank the reviewer for the great suggestion. We added statistics analysis in material and methods part. We corrected the mistakes like adding tubulin control. SAMHD1 was cloned in VR1012, so we used VR1012 as negative control. TRIM21 shRNAs were cloned in lentiviral vector pLKO.1, so we used pLKO.1 as negative control.

3. The authors use in this work two main cell lines, RD and HEK293T cells. It is very confusing when and why they use one cell line or the other. It should be explained better.

Response: We thank the reviewer for the question. RD cells contain higher TRIM21 and lower SAMHD1, while HEK293T cells contain lower TRIM21 and higher SAMHD1. So, we knock down SAMHD1 and overexpress TRIM21 in HEK293T cells and knock down TRIM21 in RD cells. For the degradation or the interaction experiments, HEK293T cells were usually used.

4. Authors could perform some of the experiments under endogenous levels of expression of the proteins. Pull down experiments in figure 2 could be done with endogenous TRIM21, at least in RD cells where they claim a major expression of the protein. The same for the microscopic images in 2I (in addition, the authors should add another irrelevant trim, as a control).

Response: As suggested by the review, we used endogenous TRIM21 antibody to detect its interaction with SAMHD1 and another TRIM family member TRIM25 as a control (Fig.2G and 2J).

5. Published reports claim that SAMHD1 does not inhibit the replication of non-retro RNA viruses (i.e. Choi J et al 2015. DOI: 10.1186/s12977-015-0174-4). Experiments such as adding SAMHD1 and then monitoring the disimintion of viral RNA/protein levels could help to strengthen their conclusions.

Response: As suggested by the review, we detected the effect on SAMHD1 on non-retro RNA viruses RSV and VSV (Fig. EV2). We observed that SAMHD1 could inhibit VSV-GFP replication within 24 h. However, accumulating viruses might evade the inhibition of SAMHD1 at 24 h, which is consistent with Choi's study. Here, we also observe that SAMHD1 couldn't inhibit RSV replication. As suggested by the reviewer, we have detected viral RNA and protein levels in Fig.1, 3 and Fig.EV1 in order to strengthen our conclusion.

6. Some publications suggest TRIM21 as an activator of innate immune responses and antiviral factor through its interaction (Liu H et al. Front immunol.2018 DOI: 10.3389/fimmu.2018.02479) with RLR signaling factors, such as RIG-I and MAVS (Xue B, et al. J Virol 2018. DOI: 10.1128/JVI.00321-18). Could the authors exclude this?

Response: We thank the reviewer for the great question. Indeed, the previous studies reported that TRIM21 is an activator or enhancers of innate immune responses [3, 6]. However, as an E3 ligase, TRIM21 has also been reported to negatively regulate the innate immune response to dsDNA by degrading DDX1, a sensor of dsDNA [4]. TRIM21 also contributed to the negative regulatory function of the Nmi-IFI35 complex on innate antiviral signaling [5]. So, TRIM21 mediates dual effector and sensor functions by facilitating simultaneous proteasomal degradation of virions and innate immune signaling [1, 2]. Therefore, the role of TRIM21 in viral infection is more complicated due to its E3 ligase function. Here, we couldn't exclude this possibility which needs to be further investigated in the future.

7. It is known that proteases in picornavirus can play an important role inhibiting key steps in the activation of innate immune responses. Can the authors exclude an effect of some viral protein in the context of infection in their work? (Wang B et al, Plos Pathog 2013. DOI: 10.1371/journal.ppat.1003231). Supplemental Figure 3E is not conclusive.

Response: We agree with the reviewer's opinion that some viral proteins encoded by some viruses are required for evading the host restriction. For EV71, we had tried to delete certain protein like 2C which resulted in non-infectious virus, so we couldn't examine the effect of some viral protein in the infection. So, we only showed that EV71 non-structural proteins couldn't downregulate SAMHD1 expression. We also modified the sentence.

8. There is the same amount of mRNA levels in both cell lines in figure 1C for SAMHD1 and other restriction factors, with and without infection. How authors explain this?

Response: We observed that the mRNA change of SAMHD1 shows no statistical significance between EV71 infection or un-infection group 72 h post infection, which also observed in Fig.EV3B. while the protein levels of SAMHD1 were obviously downregulated by TRIM21 expression. In appendix data (Figure. S1) we provided the mRNA change of diverse host factors upon infection in a time course. Although mRNA levels of A3G, Mx2 and BST-2 were upregulated at a different degree with EV71 infection, due to far lower than SAMHD1 mRNA level, we couldn't see the change in Figure.1C.

9. In figure 1E, authors should include knock down of SAMHD1 in RD cells. Are the experiments performed in 293T, or RD cells?

Response: Fig.1E was performed in 293T cells. As suggested by the reviewer, we also performed it in RD cells (Fig.1G and 1H).

10. I would suggest using another TRIM family protein as control, in the experiments done in figure 3 to show that the results are specific of TRIM21.

Response: We thank the reviewer for the suggestion. We detected the effect of TRIM25 on SAMHD1 as well as its binding to SAMHD1 (Fig.2E, 2G and 2J).

11. Why there is a difference in induction of IFN β in figure 4?

Response: Thank for the reviewer's great question. We deduced that the different sensitivity of the various tissues to EV71 infection, caused lower IFN β production in heart and lung and higher IFN β production in muscles.

Referee #3:

In this manuscript, the authors report that cellular SAMHD1 can restrict enterovirus 71 infection. Follow-up mass spec and co-IP experiments indicated that SAMHD1 interacts with TRIM21, indicating a possible functional connection between SAMHD1 and TRIM21. Consistent with the hypothesis that TRIM21 targets SAMHD1 for proteasomal degradation, cell lines and neonatal mouse tissues with lower SAMHD1 expression were more susceptible to EV71 infection.

Co-IP approaches with TRIM21 and SAMHD1 domain mutants, indicated that these proteins interact through the PRY-SPRY domain of TRIM21 and the N-terminal 1-547 aa of SAMHD1. Despite being sufficient for interaction with TRIM21, the 1-547 aa region of SAMHD1 was not degraded by TRIM21, consistent with the extreme SAMHD1 C-terminus being ubiquitinated by TRIM21. Cell-based assays with K-to-R point mutants indicated that K622 in the SAMHD1 C-terminus is ubiquitinated by TRIM21. Taken together, the authors conclude that EV71 induces TRIM21 as an antagonistic mechanism to decrease cellular SAMHD1 levels, thereby favoring its infective cycle.

1. Does this manuscript report a single key finding?

Yes; TRIM21 degrades the SAMHD1 viral restriction factor, thereby favoring EV71 infection.

2. Is the reported work of significance, or does it describe a confirmatory finding or one that has already been documented using other methods or in other organisms etc?

Yes; while it has been reported that SAMHD1 can restrict retroviruses, this work shows that unrelated RNA viruses are also restricted by SAMHD1, and that TRIM21 mechanistically controls SAMHD1 abundance.

3. Is it of general interest to the molecular biology community?

Yes; SAMHD1 is an interesting factor because of its AGS-disease association, as well as retroviral restriction. How SAMHD1 is regulated, and that it targets other viral species is in this reviewer's opinion interesting for a broad audience.

4. Is the single major finding robustly documented using independent lines of experimental evidence?

For a large part the presented data is compelling, and as a whole supports the main conclusion. However, several key experiments are either not performed or cannot be properly interpreted in their current form.

Main comments for authors:

A) The effect in EV71 infection is measured through viral RNA or VP1 protein levels. To understand if SAMHD1 and/or TRIM21 levels influence infection in a biological meaningful way, at least one key experiment should include a measurement of infectious virus.

Response: We thank the reviewer for the suggestion. We had detected viral RNA or VP1 protein levels to measure EV71 replication in this manuscript, for example, Fig.1, 3, Fig. EV1.

B) The presented data reasonably supports that TRIM21 and SAMHD1 individually influence EV71 infection, and that they interact. However, in this reviewer's opinion, genetic epistasis evidence (e.g. Fig. 3H, 6-G-I) is not very strong.

Response: We thank the reviewer for the suggestion. In current version, the expression of SAMHD1 and TRIM21 in RD or HEK293T cells (Fig.2H, 2I, 3 and EV1) or the diverse tissues of mice (Fig.4) all supported that TRIM21-mediated the degradation of SAMHD1 affects SMHD1 inhibition on EV71 replication. Overexpression of TRIM21 in SAMHD1 knockdown cells had no effect on EV71 infection, further supporting their correlation (Fig.3F).

C) Given the plethora of cellular roles described for both TRIM21 and SAMHD1, it is unclear whether their manipulation affects the general fitness of the cell to support viral replication, or affects specifically EV71.

Response: We thank the reviewer for the great question. We observed that the proliferation of cells was not affected by TRIM21 or SAMHD1 knockdown or overexpression.

D) The conclusion that EV71 has developed IFN-dependent TRIM21 induction as an antagonism strategy to degrade SAMHD1 seems far-fetched in this reviewer's opinion. The general thought is that type I IFNs are the key antiviral response for practically all viruses, and that virtually all viruses induce type I IFNs during their life cycle, yet have developed viral antagonists to limit this. In this reviewer's opinion, it seems much more likely that TRIM21 would be a cellular negative feedback mechanism to limit SAMHD1 levels after type I IFN exposure. Since EV71 (as most viruses) induces IFN β , it also induces this response.

Response: We thank the reviewer for the great suggestion. As suggested by the reviewer, we modified the sentence and discussed it in discussion part.

E) In this reviewer's opinion, cannot be interpreted for the following reasons:

- There are no MW markers, making it unclear whether the IPed SAMHD1 bands indeed exclusively are of SAMHD1+at least 8.5 kDa (for Ub).

Response: We thank the reviewer for the suggestion. We redid these experiments with no-tagged TRIM21 and labeled the MW markers in Ub data (Fig.8C and 8D).

- Ubiquitination can only be shown if one pulls on Ub and blots for SAMHD1, or inverse. Here the IP is for SAMHD1 and the WB as well.

Response: We agree with the reviewer's suggestion, we actually blotted it for Ub with anti-flag antibody, we corrected the label.

- The Ub and TRIM21 constructs have both the same tag, there should be a dark smear of ubiquitinated cellular proteins in the TRIM21 input (which is not there).

Response: We thank the reviewer for the suggestion. We redid these experiments with no tagged TRIM21 which is more reasonable for this experiment (Fig.8C and 8D). For last version, TRIM21 is around 52KDa, while SAMHD1 is around 72KDa. The band was cut off and stained, so we didn't see a dark smear band.

- The "ubiquitin-flag" construct is not described in the manuscript, but the name suggests that it may have a C-terminal tag (which would be unable to conjugate, and could explain the lack of protein smear as pointed out in the point above).

Response: We thank the reviewer for the great suggestion. We described it in material and methods part.

In order to fully substantiate the claims made in the manuscript, this reviewer's opinion the following experiments should be included (letters correspond to main comments above):

A+B): Epistasis experiment in which Trim21 and Samhd1 are individually depleted, as well as in combination, followed by EV71 infection, with sampling of produced virus in a growth curve (read out by plaque assay or TCID50). Samhd1 knock-down seems efficient, but for Trim21, a CRISPR knock-out line (or MEFs from Trim21^{-/-} mice) would be suggested.

Response: We constructed CRISPR-Cas9-TRIM21 in RD cells and examined the effect on SAMHD1 and EV71 infection (Fig.3). We hope we can examine the effect in Trim21^{-/-} mice in the future

C): Determine the effect of Samhd1 and Trim21 knock-down and over-expression on multi-cycle infection with an unrelated virus (e.g. VSV).

Response: We thank the reviewer for the suggestion. We investigated the effect of SAMHD1 on RSV and VSV viruses (Fig.EV2).

D): To get an indication if TRIM21 upregulation is a viral antagonism strategy, the authors could determine the effect of ablating the IFN-response (Ifnar^{-/-}) on viral titers, compared to Samhd1^{-/-} and restoration of Samhd1 in the Ifnar^{-/-} background. It stands to reason that the strongest effector has provided the strongest evolutionary pressure to be antagonized.

Response: We thank the reviewer for the great suggestion. We checked the effect of the depletion of IFN receptor on EV71 infection and found slightly increased EV71 infection due to SAMHD1, while knockdown of IFNAR and SAMHD1 both resulting in moderately increased EV71 infection (Fig.EV3E).

E): A His-Ub denaturing pull-down (urea; with Ni-NTA), followed by a blot for SAMHD1 should be performed. Blots of the input and PD for His-Ub and SamHD1-HA should be shown. MW size markers should be carefully indicated (as well as in the rest of the figures); there should be no unmodified SAMHD1 in the pull-down.

Response: We thank the reviewer for the great suggestion. We did the experiment and demonstrated the direct interaction between TRIM21 and SAMHD1 (Fig.7G).

E2): Given the moderately effective knock-down of Trim21, and mild effects on SAMHD1 expression, a Trim21 CRISPR knock-out line in which SAMHD1 stability is measured (compared to non-targeted cells) in the absence and presence of MG132, should be included.

Response: We thank the reviewer for the great suggestion. We constructed CRISPR-Cas9- TRIM21 and examined the effect on SAMHD1 and EV71 infection (Fig.3).

Other comments:

1) Line 141: "nucleUS to nuclear membrane"?

Response: We corrected the mistake.

2) Line 200: "By" Delete word?

Response: We corrected the mistake.

3) Lines 230-235: To the best of my knowledge HEK-293Ts do not have a functional cGAS-STING pathway.

Response: We thank the reviewer for the remind. For this experiment, we detected the IFN production when overexpression of TRIM21 in the presence of genotoxic agent hydroxyurea (HU), according to the protocol of recent report [7]. We indeed observed the effect of TRIM21 on SAMHD1-associated IFN induction.

4) Alignment issue in figure panel 6A.

Response: Thank the reviewer for the careful observation, we corrected the mistake.

5) The interpretation of Fig. 6G-I is unclear. My interpretation is that TRIM21-HA expression and Samhd1 knock-down each equally increase the reporter. Yet, deleting Samhd1 in the TRIM21-HA expressing cells does not alter the reporter induction, suggesting that the induction by TRIM21-HA is independent of Samhd1.

Response: We thank the reviewer for the great suggestion. We modified the interpretation according to the suggestion in the manuscript.

6) Fig 7C: Unclear why there is no decrease in SAMHD1 when TRIM21 is expressed.

Response: In order to make SAMHD1 express well, the cells were treated with 10 μ M MG132 (proteasome inhibitor) to avoid SAMHD1 to be degraded.

7) Fig 7F: The labeling for the top plasmid is missing.

Response: We added the label.

8) The neonatal mouse figure seems a bit out of place at the current position, since it essentially demonstrates that there is altered Samhd1/Trim21 expression at the sites of infection. As there is not genetics, and contribution to causality, it may better fit earlier in the manuscript (e.g. with Fig. 1) to make a case for correlation between the two factors and infection susceptibility.

Response: We thank the reviewer for the suggestion. In Fig.2 and Fig.3, we identified TRIM21 as an important regulator of SAMHD1 by mass spectrum first, then investigated the correlation between the two factors and infection susceptibility. So, we have to put it in Fig.4.

9) Line 310/311: Without cell-free in vitro reconstitution it cannot be claimed that TRIM21 directly targets SAMHD1 for degradation.

Response: We thank the reviewer for the suggestion. We demonstrated the direct interaction of TRIM21 and SAMHD1 expressed from *E.coli* by positive and reverse pull down experiments (Fig.7G).

10) Line 378-381: Not entirely clear what is meant here. TRIM-Away is based on intra-cellular antibody-directed targeting of TRIIM21 to the target of the antibody. Since this does not alter TRIM21 levels itself, it is difficult to see why this would cause off-targets to endogenous TRIM21 targets such as SAMHD1.

Response: We thank the reviewer for the remind. We modified the sentence. TRIM-Away needs the introduction of exogenous TRIM21, so it should alter TRIM21 level [8]. Our initial opinion is that TRIM21 might recognize its other substrates and induce them degradation by its activity of E3 ligase, like SAMHD1.

References:

1. Bottermann M, James LC (2018) Intracellular Antiviral Immunity. *Adv Virus Res* **100**: 309-354
2. Vaysburd M, Watkinson RE, Cooper H, Reed M, O'Connell K, Smith J, Cruickshanks J, James LC (2013) Intracellular antibody receptor TRIM21 prevents fatal viral infection. *Proceedings of the National Academy of Sciences of the United States of America* **110**: 12397-401
3. Versteeg GA, Rajsbaum R, Sanchez-Aparicio MT, Maestre AM, Valdiviezo J, Shi M, Inn KS, Fernandez-Sesma A, Jung J, Garcia-Sastre A (2013) The E3-ligase TRIM family of proteins regulates signaling pathways triggered by innate immune pattern-recognition receptors. *Immunity* **38**: 384-98
4. Zhang Z, Bao M, Lu N, Weng L, Yuan B, Liu YJ (2013) The E3 ubiquitin ligase TRIM21 negatively regulates the innate immune response to intracellular double-stranded DNA. *Nature immunology* **14**: 172-8
5. Das A, Dinh PX, Pattnaik AK (2015) Trim21 regulates Nmi-IFI35 complex-mediated inhibition of innate antiviral response. *Virology* **485**: 383-92
6. Liu H, Li M, Song Y, Xu W (2018) TRIM21 Restricts Coxsackievirus B3 Replication, Cardiac and Pancreatic Injury via Interacting With MAVS and Positively Regulating IRF3-Mediated Type-I Interferon Production. *Front Immunol* **9**: 2479
7. Coquel F, Silva MJ, Techer H, Zadorozhny K, Sharma S, Nieminuszczy J, Mettling C, Dardillac E, Barthe A, Schmitz AL, et al. (2018) SAMHD1 acts at stalled replication forks to prevent interferon induction. *Nature* **557**: 57-61
8. Clift D, McEwan WA, Labzin LI, Konieczny V, Mogessie B, James LC, Schuh M (2017) A Method for the Acute and Rapid Degradation of Endogenous Proteins. *Cell* **171**: 1692-1706.e18

Thank you for the submission of your revised manuscript to EMBO reports. We have now received the comments from the three referees that were asked to re-assess it, which you will find below.

I am sorry to say that the evaluation of your revised manuscript is not a positive one. As you will see, two referees (#1 and #3) state, although several of their points have been addressed, several issues have not been resolved. Referee #2 is more positive, but also has concerns regarding a major conclusion of the study.

Given these substantial remaining concerns, the fact that you already had a chance to significantly revise the study, and that we allow a single round of revision only, I am afraid that we cannot offer to publish the manuscript at this point. I am sorry that this decision emerges as the outcome of a lengthy review process, but given that the referees are still not convinced by the current set of data, I have no other option but to reject your manuscript.

I am sorry that I could not bring better news this time and hope that the referee comments will be helpful in your continued work in this area.

REFeree REPORTS

Referee #1:

My fundamental concern with the data is the unexpected finding that TRIM21 binds SAMHD1. Given that this is contrary to the reported binding functions of TRIM21 (which are supported by structures and detailed binding data) the bar to support such a claim must necessarily be high. I had therefore requested equivalent *in vitro* binding experiments by ITC or SPR or similar to confirm TRIM21: SAMHD1 interaction. As both TRIM21 and SAMHD1 proteins are known to be well expressed and easy to work with (there are x-ray structures of both) this is not an unreasonable request. However, this has not been done and so my concern remains that the pull-down data reflects an indirect association or artefact of this approach, something that is not uncommon.

Referee #2:

The authors have done a good job in adding new experimental evidence that strengthen their findings, that is, that IFN-inducible TRIM21 negatively regulates the host antiviral response by promoting degradation of SAMHD1, a host factor known to reduce replication of several viruses, as also shown in this paper for new viruses not tested before. My only remaining concern is that the authors keep concluding that EV71 (and perhaps other viruses) induce IFN to upregulate TRIM21 so SAMHD1 becomes degraded therefore promoting viral replication. However, the authors have not proven that this is in fact a new viral evasion mechanism. Rather, the results can be explained by a natural negative feed-back of the IFN response that prevents excessive IFN responses. Obviously, if one eliminates negative feed-backs of the IFN response, one promotes also virus replication, but this does not qualify these negative feed-back pathways as mechanisms of viral evasion. Thus, all references in the manuscript that suggest that TRM25 induction by IFN is a novel viral evasion mechanism should be eliminated.

Referee #3:

The authors submitted a revised version of the manuscript with new experimental data that partially address my previous concerns. It is still my opinion that the authors provide an impressive amount of data, and report an interesting finding. However, several of the original concerns were either not adequately addressed, or not addressed at all.

Comments from this reviewer pertaining to the revised version are indicated with ">>Reviewer3:".

A+B): Epistasis experiment in which Trim21 and Samhd1 are individually depleted, as well as in combination, followed by EV71 infection, with sampling of produced virus in a growth curve (read out by plaque assay or TCID50). Samhd1 knock-down seems efficient, but for Trim21, a CRISPR knock-out line (or MEFs from Trim21^{-/-} mice) would be suggested.

Response: We constructed CRISPR-Cas9-TRIM21 in RD cells and examined the effect on SAMHD1 and EV71 infection (Fig.3). We hope we can examine the effect in Trim21^{-/-} mice in the future

>>Reviewer3: I cannot find any new data measuring the effect on infection EV71 virus by TCID50 or plaque assay has been provided. Although it is clear that SAMHD1 affects viral RNA and protein, my original point stands: it is currently unclear whether this is biologically meaningful. It may be that there is only a minor fold difference in infectious viral titer.

Epistasis: While the authors introduce new data from TRIM21 knock-out cells which more robustly reproduce what they had previously shown using shRNAs, they did not introduce the epistasis setup that I previously suggested: compare shSAMHD1 vs. sgRNA TRIM21 vs. (shSAMHD1+sgTRIM21) by measuring infectious virus output by TCID50. In my opinion, (although there is quite a bit of data presented indicating a functional correlation between both, and direct physical interaction) there still is no experiment data that clearly indicates this. It is clear that all reagents and tools are present for this one experiment that would have addressed both listed points, yet there is not argument why this was not provided.

C): Determine the effect of Samhd1 and Trim21 knock-down and over-expression on multi-cycle infection with an unrelated virus (e.g. VSV).

Response: We thank the reviewer for the suggestion. We investigated the effect of SAMHD1 on RSV and VSV viruses (Fig.EV2).

>>Reviewer3: This point has been sufficiently addressed by RSV and VSV infections.

D): To get an indication if TRIM21 upregulation is a viral antagonism strategy, the authors could determine the effect of ablating the IFN-response (Ifnar^{-/-}) on viral titers, compared to Samhd1^{-/-} and restoration of Samhd1 in the Ifnar^{-/-} background. It stands to reason that the strongest effector has provided the strongest evolutionary pressure to be antagonized.

Response: We thank the reviewer for the great suggestion. We checked the effect of the depletion of IFN receptor on EV71 infection and found slightly increased EV71 infection due to SAMHD1, while knockdown of IFNAR and SAMHD1 both resulting in moderately increased EV71 infection (Fig.EV3E).

>>Reviewer3: New data provided sufficiently address this point.

E): A His-Ub denaturing pull-down (urea; with Ni-NTA), followed by a blot for SAMHD1 should be performed. Blots of the input and PD for His-Ub and SamHD1-HA should be shown. MW size markers should be carefully indicated (as well as in the rest of the figures); there should be no unmodified SAMHD1 in the pull-down.

Response: We thank the reviewer for the great suggestion. We did the experiment and demonstrated the direct interaction between TRIM21 and SAMHD1 (Fig.7G).

>>Reviewer3: While the direct interaction between TRIM21 and SAMHD1 does not pertain to this, the authors did provide new IPs addressing SAMHD1 ubiquitination (Figures 8C and 8D). Especially for figure 8C I still have the same concerns: based on the MW it seems that after SAMDH1 IP and blotting for ubiquitin, there is still unmodified SAMHD1 present. What makes it in my opinion impossible to interpret is that standard controls are lacking: there is no "no-bait" control and the flag-input is not shown. It is not unlikely that TRIM21 overexpression simply increases poly-ubiquitin levels in the cells and these are non-specifically IPed. Likewise, for 8D not knowing whether flag-ubiquitin levels are equal in the input makes interpretation difficult.

E2): Given the moderately effective knock-down of Trim21, and mild effects on SAMHD1 expression, a Trim21 CRISPR knock-out line in which SAMHD1 stability is measured (compared to non-targeted cells) in the absence and presence of MG132, should be included.

Response: We thank the reviewer for the great suggestion. We constructed CRISPR-Cas9- TRIM21 and examined the effect on SAMHD1 and EV71 infection (Fig.3).

>>Reviewer3: Indeed, the authors provided data from TRIM21 knock-out cells, which phenocopy what they had previously shown with knock-down. However, my point whether TRIM21 deletion changes proteasome-dependent stability of SAMHD1 was not addressed. No straight-forward cycloheximide and MG132/epoxomicin treatments were performed.

Resubmission

7 August 2019

Referee #1:

My fundamental concern with the data is the unexpected finding that TRIM21 binds SAMHD1. Given that this is contrary to the reported binding functions of TRIM21 (which are supported by structures and detailed binding data) the bar to support such a claim must necessary be high. I had therefore requested equivalent vitro binding experiments by ITC or SPR or similar to confirm TRIM21:SAMHD1 interaction. As both TRIM21 and SAMHD1 proteins are known to be well expressed and easy to work with (there are x-ray structures of both) this is not an unreasonable request. However, this has not been done and so my concern remains that the pull-down data reflects an indirect association or artefact of this approach, something that is not uncommon.

Response : Thank the reviewer for the great suggestion. In order to confirm and verify the direct interaction between SAMHD1 and TRIM21 *in vitro*, we have performed Fluorescence resonance energy transfer (FRET) assay and microscale thermophoresis (MST) assay. As shown in Fig. 8 of the revised manuscript, TRIM21 was observed to interact with SAMHD1 directly. Besides, we have also made our effort to perform the ITC assay. Given that high concentration (> 500µM) of protein and the absence of glycerin are required by ITC assay, we have tried to purify and concentrate SAMHD1 and TRIM21 to meet this criteria. However, SAMHD1 began to precipitate after purification and concentration to 50µM without the MgCl₂, dNTPs and glycerin, which were reported to stabilize SAMHD1 in structure research [1]. Thus, we have to give up the ITC assay.

Referee #2:

The authors have done a good job in adding new experimental evidence that strengthen their findings, that is, that IFN-inducible TRIM21 negatively regulates the host antiviral response by promoting degradation of SAMHD1, a host factor known to reduce replication of several viruses, as also shown in this paper for new viruses not tested before. My only remaining concern is that the authors keep concluding that EV71 (and perhaps other viruses) induce IFN to upregulate TRIM21 so SAMHD1 becomes degraded therefore promoting viral replication. However, the authors have not proven that this is in fact a new viral evasion mechanism. Rather, the results can be explained by a natural negative feed-back of the IFN response that prevents excessive IFN responses. Obviously, if one eliminates negative feed-backs of the IFN response, one promotes also virus replication, but this does not qualify these negative feed-back pathways as mechanisms of viral evasion. Thus, all references in the manuscript that suggest that TRM25 induction by IFN is a novel viral evasion mechanism should be eliminated.

Response: Thanks to the reviewer for the insightful suggestion, and we have modified our conclusions as suggested by the reviewer in the revised manuscript, such as lanes 197, 335, 358.

Referee #3:

The authors submitted a revised version of the manuscript with new experimental data that partially address my previous concerns. It is still my opinion that the authors provide an impressive amount of data, and report an interesting finding. However, several of the original concerns were either not adequately addressed, or not addressed at all.

Comments from this reviewer pertaining to the revised version are indicated with ">>Reviewer3:".

A+B): Epistasis experiment in which Trim21 and Samhd1 are individually depleted, as well as in combination, followed by EV71 infection, with sampling of produced virus in a growth curve (read out by plaque assay or TCID50). Samhd1 knock-down seems efficient, but for TRIM21, a CRISPR knock-out line (or MEFs from TRIM21^{-/-} mice) would be suggested.

Response: We constructed CRISPR-Cas9-TRIM21 in RD cells and examined the effect on SAMHD1 and EV71 infection (Fig.3). We hope we can examine the effect in TRIM21^{-/-} mice in the future

>>Reviewer3: I cannot find any new data measuring the effect on infection EV71 virus by TCID50 or plaque assay has been provided. Although it is clear that SAMHD1 affects viral RNA and protein, my original point stands: it is currently unclear whether this is biologically meaningful. It may be that there is only a minor fold difference in infectious viral titer.

Response: We are grateful to the reviewer for the valuable advice, and we have measured the TCID50 of EV71 in supernatants (Fig. 1G, 1J, 1M, 3D and 3H).

Epistasis: While the authors introduce new data from TRIM21 knock-out cells which more robustly reproduce what they had previously shown using shRNAs, they did not introduce the epistasis setup that I previously suggested: compare shSAMHD1 vs. sgRNA TRIM21 vs. (shSAMHD1+sgTRIM21) by measuring infectious virus output by TCID50. In my opinion, (although there is quite a bit of data presented indicating a functional correlation between both, and direct physical interaction) there still is no experiment data that clearly indicates this. It is clear that all reagents and tools are present for this one experiment that would have addressed both listed points, yet there is not argument why this was not provided.

Response: Thanks to the reviewer's suggestion. We detected EV71 replication in RD cells whose TRIM21 and SAMHD1 are individually depleted, as well as in combination. We found that increased SAMHD1 resulting from TRIM21 deletion decreased EV71 replication, whereas siSAMHD1+sgTRIM21 lost the ability to inhibit EV71 replication, indicating that EV71 replication was negatively correlated with SAMHD1 level (Fig. EV2A and B).

E): A His-Ub denaturing pull-down (urea; with Ni-NTA), followed by a blot for SAMHD1 should be performed. Blots of the input and PD for His-Ub and SAMHD1-HA should be shown. MW size markers should be carefully indicated (as well as in the rest of the figures); there should be no unmodified SAMHD1 in the pull-down.

Response: We thank the reviewer for the great suggestion. We did the experiment and demonstrated the direct interaction between TRIM21 and SAMHD1 (Fig.7G).

>>Reviewer3: While the direct interaction between TRIM21 and SAMHD1 does not pertain to this, the authors did provide new IPs addressing SAMHD1 ubiquitination (Figures 8C and 8D). Especially for figure 8C I still have the same concerns: based on the MW it seems that after SAMHD1 IP and blotting for ubiquitin, there is still unmodified SAMHD1 present. What makes it in my opinion impossible to interpret is that standard controls are lacking: there is no "no-bait" control and the flag-input is not shown. It is not unlikely that TRIM21 overexpression simply increases poly-ubiquitin levels in the cells and these are non-specifically IPed. Likewise, for 8D not knowing whether flag-ubiquitin levels are equal in the input makes interpretation difficult.

Response: Be grateful for the reviewer's suggestion. We re-performed the Co-IP assay in this revised manuscript. Here, we added "no-bait" control, stained flag-ubiquitin in the input. The results were shown in Fig. 8C and 8D and suggested that TRIM21 increased poly-ubiquitin levels of SAMHD1.

E2): Given the moderately effective knock-down of TRIM21, and mild effects on SAMHD1 expression, a TRIM21 CRISPR knock-out line in which SAMHD1 stability is measured (compared to non-targeted cells) in the absence and presence of MG132, should be included.

Response: We thank the reviewer for the great suggestion. We constructed CRISPR-Cas9- TRIM21 and examined the effect on SAMHD1 and EV71 infection (Fig.3).

>>Reviewer3: Indeed, the authors provided data from TRIM21 knock-out cells, which phenocopy what they had previously shown with knock-down. However, my point whether TRIM21 deletion changes proteasome-dependent stability of SAMHD1 was not addressed. No straight-forward cycloheximide and MG132/epoxomicin treatments were performed.

Response: As suggested by the reviewer, we conducted this experiment in revised manuscript and found that MG132 had no effect on SAMHD1 stability when TRIM21 was deleted (Fig. EV2C).

Reference

1. Zhu C, Gao W, Zhao K, Qin X, Zhang Y, Peng X, Zhang L, Dong Y, Zhang W, Li P, *et al.* (2013) Structural insight into dGTP-dependent activation of tetrameric SAMHD1 deoxynucleoside triphosphate triphosphohydrolase. *Nat Commun* 4: 2722

3rd Editorial Decision

18 September 2019

Thank you for the submission of your revised manuscript to our editorial offices. We have now received the reports from the two referees that were asked to re-evaluate your study, you will find below. As you will see, referee #1 has remaining concerns and suggestions I ask you to address in a final revised version of your manuscript. Please clearly indicate number of replicates and their type (biological or technical).

Further, I have these editorial requests:

- Please provide the abstract written in present tense.
- We allow up to 5 EV figures, and in order to keep the paper compact, it would be great to have not more than 8 main figures. You have submitted 6 EV figures, and 9 main figures. Please re-arrange this in order to have not more than 5 EV figures and 8 main figures. In case, additional data can be moved to the Appendix. Please follow the nomenclature Appendix Figure Sx, Appendix Table Sx etc. throughout the text, and also label the figures and tables according to this nomenclature. Please take care to update the callouts for all figure panels in the final version.

See also our guide for figure preparation:

- There is a call out for Table 1 in the manuscript text, but no Table 1 was uploaded. Please do this.
- Please add scale bars to all microscopic images. Please refrain from any writing indicating their size directly on the bars in the images. Please indicate the size only in the respective figure legend.
- Please add page numbers to the Appendix and the Appendix TOC.
- We require that primary datasets produced in this study (e.g. mass spec. data) are deposited in an appropriate public database. See: <http://www.embopress.org/page/journal/14693178/authorguide#datadeposition>

The accession numbers and database should be listed in a formal "Data Availability" section (placed after Materials & Methods) that follows the model below. Please do that for your manuscript. Please note that the Data Availability Section is restricted to new primary data that are part of this study.

Data availability

- Several WB panels are over-contrasted. Moreover, panels in one figure often show very different contrast and brightness. Please show the WBs with panels with equal contrast, and as unmodified as possible.

- For the V3 of this manuscript you provided source data for the Western blots. This is highly appreciated. Please also do this for the final revised version. We require source data (full scans of Western blots with size markers) for the main figures, the EV figures and the Appendix. For main and EV figures, please combine the source data from one figure into one pdf file. Do not combine source data from different figures in one file. The source data will be linked in the online version of the paper to each main and EV figure. The source data for the Appendix figures can be combined in one file, though.

- Please make sure that, where applicable, the number "n" for how many independent experiments (biological replicates or technical replicates) were performed, the bars and error bars (e.g. SEM, SD) and the test used to calculate p-values in the respective figure legends is indicated. Please provide statistical testing where applicable, and also add a paragraph detailing this to the methods section. See:

<http://www.embopress.org/page/journal/14693178/authorguide#statisticalanalysis>

- Please have the manuscript carefully proofread by a native speaker. There are still some grammatical errors present. Our publisher also offers a manuscript editing service: <https://wileyeditingservices.com/en/english-language-editing/>

- Finally, please find attached a word file of the manuscript text (provided by our publisher) with changes we ask you to include in your final manuscript text, and some queries, we ask you to address. Please provide your final manuscript file with track changes, in order that we can see the modifications done.

In addition I would need from you:

- a short, two-sentence summary of the manuscript
- two to three bullet points highlighting the key findings of your study
- a schematic summary figure (in jpeg or tiff format with the exact width of 550 pixels and a height of not more than 400 pixels) that can be used as a visual synopsis on our website.

REFEREE REPORTS

Referee #1:

The authors have performed binding experiments using MST to measure TRIM21/SAMHD1 interaction. I'm not able to determine whether there are three biological replicates of the data; it may be a single serial dilution to measure three times or measuring the same capillaries three times.

The y-axis is weirdly labelled; it should be ratio of fluorescence (before heating / after heating) X100. They do not specify at what time they evaluated this in the methods. They also do not

indicate which capillary type measurements were made in nor the buffer used nor whether they added any tween to avoid surface adsorption.

The Kd is quoted with insane precision; it should be 270 nM.

Referee #3:

Point 1: Measurement of TCID50.

The authors now provide data indicating that viral titers decrease about 1 log upon Trim21 ablation, and increase by about 1 log upon Samhd1 ablation. Together, these results sufficiently address my previous concerns, and indicate that both factors significantly influence infectious virus output.

Point 2: Epistasis experiment of simultaneous SAMHD1 and TRIM21 ablation.

The authors have added new evidence in Fig. EV2B that demonstrates that TRIM21 indeed likely inhibits SAMHD1 as claimed, since TRIM21 ablation in SAMHD1 knock-down cells no longer decreases virus titers. The authors have sufficiently addressed my previous concerns on this point.

Point 3: SAMHD1 ubiquitination by TRIM21

The authors have included no-bait controls and loading controls for ubiquitin. The new results suggest that SAMHD1-associated ubiquitin indeed increases if TRIM21 and K48 ubiquitin are expressed, but not with K63 ubiquitin, or in a K622R SAMHD1 mutant. Together, these data sufficiently address my previous comments.

Point 4: CHX chase for SAMHD1 stability in TRIM21-deficient cells.

The authors included new convincing data that SAMHD1 is no longer unstable in the absence of TRIM21, or in the absence of proteasomal degradation. These results sufficiently address my previous comments.

2nd Revision - authors' response

9 October 2019

Referee #1:

The authors have performed binding experiments using MST to measure TRIM21/SAMHD1 interaction. I'm not able to determine whether there are three biological replicates of the data; it may be a single serial dilution to measure three times or measuring the same capillaries three times.

The y-axis is weirdly labelled; it should be ratio of fluorescence (before heating / after heating) X100. They do not specify at what time they evaluated this in the methods. They also do not indicate which capillary type measurements were made in nor the buffer used nor whether they added any tween to avoid surface adsorption.

The Kd is quoted with insane precision; should be 270 nM.

Response:

Thanks for the referee's suggestion. A two-fold dilution series ranging from 0.1 nM to 3 μ M TRIM21 was prepared in PBS (pH 7.4) with 0.05% (v/v) Tween. The experiments were repeated three times using different dilution and the data was a representative of three independent experiments using a single serial dilution and analyzed using the NanoTemper® analysis software (F Norm = F Hot / F Cold). Measurements were performed at 21.5 {degree sign}C using 40% MST power with laser on/off times of -1 s/0 s and 19 s/20 s, respectively. The capillary we used is typed as K002. We also described above details in revised manuscript.

As suggested, the Kd is now quoted as 270±92 nM.

Corresponding Author Name: Wenyang Zhang

Manuscript Number: EMBOR-2018-47528